



# Rarefied particle motions on hillslopes: 3. Entropy

David Jon Furbish[1], Sarah G. W. Williams[1], and Tyler H. Doane[2, 3]

[1]Department of Earth and Environmental Sciences, Vanderbilt University, Nashville, Tennessee, USA
[2]Department of Geosciences, University of Arizona, Tucson, Arizona, USA
[3]Current: Department of Earth and Atmospheric Sciences, Indiana University, Bloomington, Indiana, USA

**Correspondence:** David Furbish (david.j.furbish@vanderbilt.edu)

**Abstract.** Theoretical and experimental work (Furbish et al., 2020a, 2020b) indicates that the travel distances of rarefied particle motions on rough hillslope surfaces are described by a generalized Pareto distribution. The form of this distribution varies with the balance between gravitational heating, due to conversion of potential to kinetic energy, and frictional cooling, due to particle-surface collisions; it varies from a bounded form associated with rapid thermal collapse to an exponential form representing isothermal conditions to a heavy-tailed form associated with net heating of particles. The generalized Pareto distribution in this problem is a maximum entropy distribution constrained by a fixed energetic "cost" — the total cumulative energy extracted by collisional friction per unit kinetic energy available during particle motions. That is, among all possible accessible microstates — the many different ways to arrange a great number of particles into distance states where each arrangement satisfies the same fixed total energetic cost — the generalized Pareto distribution represents the most probable arrangement. Because this idea applies equally to the accessible microstates associated with net cooling, isothermal conditions and net heating, the fixed energetic cost provides a unifying interpretation for these distinctive behaviors, including the abrupt transition in the form of the generalized Pareto distribution in crossing isothermal conditions. The analysis therefore represents a novel generalization of an energy-based constraint in using the maximum entropy method to infer non-exponential distributions of particle motions. Moreover, the energetic costs of individual particle motions follow an extreme-value distribution that is heavy-tailed for net cooling and light-tailed for net heating. The relative contribution of different travel distances to the total energetic cost is reflected by the product of the travel distance distribution and the cost of individual particle motions — effectively a frequency-magnitude product.

## 1 Introduction

In two companion papers (Furbish et al., 2020a, 2010b) we examine a theoretical formulation of the probabilistic physics of rarefied particle motions and deposition on rough hillslope surfaces. The formulation is based on a description of the kinetic energy balance of a cohort of particles treated as a rarefied granular gas and a description of particle deposition that depends on the energy state of the particles. The formulation predicts a generalized Pareto distribution of particle travel distances whose form varies with the balance between gravitational heating, due to conversion of potential to kinetic energy, and frictional cooling, due to particle-surface collisions. Specifically, the generalized Pareto distribution varies from a bounded form associated with thermal collapse and rapid deposition to an exponential form representing isothermal conditions to a heavy-tailed form



associated with net heating of particles and decreased deposition. The transition to a heavy-tailed form likely involves an in-
creasing conversion of translational to rotational kinetic energy leading to larger travel distances with decreasing effectiveness
of collisional friction. As described in Furbish et al. (2020b), these varying forms of the generalized Pareto distribution are
consistent with laboratory measurements of particle travel distances reported by Gabet and Mendoza (2012) and Furbish et al.
(2020b), and with field-based measurements of travel distances reported by DiBiase et al. (2017) and Roth et al. (2020).

Here we highlight a key point in Furbish et al. (2020a). Namely, the generalized Pareto distribution is not selected in an
empirical manner based on goodness-of-fit criteria applied to data sets. Rather, this distribution is dictated by the physics
of the problem, just as, for example, the Boltzmann distribution (an exponential distribution) emerges in classical statistical
mechanics from consideration of the accessible energy microstates of a gas system. In this problem the versatile form of the
generalized Pareto distribution — specifically its apparent success in describing three distinctive energetic behaviors of rarefied
particle motions — is enigmatic. Although the different energetic behaviors have a clear mechanical explanation, the transition
from a bounded form to a heavy-tailed form in crossing isothermal conditions is abrupt. The basis of this transition, including
the upper bound on travel distances prior to transition, is unclear — whether it represents a fundamental change in mechanical
behavior or is simply a mathematical curiosity of the generalized Pareto distribution.

The purpose of this third companion paper therefore is to further elaborate the probabilistic physics of particle motions as
represented by the generalized Pareto distribution. To do this we appeal to the principle of maximum entropy as outlined in the
pioneering work of Jaynes (1957a, 1957b). We specifically demonstrate that in this problem the generalized Pareto distribution
is a maximum entropy distribution constrained by a fixed total energetic "cost" — the total cumulative energy extracted by
collisional friction per unit kinetic energy available during particle motions. The relative energetic cost locally increases with
increasing travel distance for net particle cooling and rapid thermal collapse, it is uniform for isothermal conditions, and it
decreases with increasing travel distance for net particle heating. The cumulative cost involves integrating the local cost over
the particle travel distance, and the total cumulative cost is then obtained by summing over all particles. This fixed total cost
unifies the interpretation of the three energetic behaviors, where the upper bound on travel distances prior to transition is a
probabilistic mechanical outcome.

As a point of reference, the canonical example of a maximum entropy distribution is the Boltzmann distribution of the energy
states of the particles composing an ordinary gas at thermal equilibrium. Similarly, the Maxwell-Boltzmann distribution of
particle speeds, which is derived from the Boltzmann distribution, is a maximum entropy distribution. Here we are referring to
the Gibbs entropy of statistical mechanics. A maximum entropy distribution then is the unique distribution that maximizes the
Gibbs entropy, subject to constraints imposed on the system. In the canonical case these constraints consist of a fixed number
of particles and a fixed total energy, which together guarantee a fixed average energy equal to $k_B T$, where $k_B$ is the Boltzmann
constant and $T$ is temperature. Moreover, any other distribution of particle energy states satisfying these constraints would
coincide with a lower Gibbs entropy.

Jaynes (1957a, 1957b) elaborated the significance of the fact that the Gibbs entropy in statistical mechanics and the Shannon
entropy in information theory are essentially one and the same, differing only by a constant. This similarity inspired Jaynes to
champion the use of a maximum entropy criterion in choosing a probability distribution, leading to what is now known as the





maximum entropy method (aka MaxEnt or MEM). The key idea of the maximum entropy method, whether viewed as a method of statistical mechanics or as one of inferential statistics, is that it provides an unbiased choice of a distribution by honoring only what is known mechanically about a system. That is, this unbiased choice is a maximally noncommittal choice that is faithful to what we do not know; it is therefore the most reasonable choice in the absence of additional information (Jaynes, 1957a; Williamson, 2010, pp. 25 and 51). Importantly, mechanical constraints imposed on the system are part of the choice

of the distribution, as opposed to empirical fitting without regard to such constraints. The maximum entropy method has been applied in a remarkable variety of fields (Shore and Johnson, 1980; Ramirez and Carta, 2006; Verkley and Lynch, 2009; Singh, 2011; Peterson et al., 2013), including sediment transport (Furbish and Schmeeckle, 2013; Furbish et al., 2016).

In using the maximum entropy method, constraints imposed on the system normally translate to constraints imposed on the moments of the distribution. In this case the method leads to a distribution that is among the exponential family (e.g.,

exponential, Gaussian). However, applications of the maximum entropy method to non-exponential distributions, including heavy-tailed distributions, are of particular interest in many problems (Peterson et al., 2013). As described below, applying this method to heavy-tailed distributions presents a special challenge in that the first or second moment, or both of these moments, may be undefined for such distributions, including the generalized Pareto distribution (Pickands, 1975; Hosking and Wallis, 1987).

In Section 2 we provide background material, namely, the essential elements of the formulation of Furbish et al. (2020a) leading to the generalized Pareto distribution of particle travel distances, and a summary of the properties and derivation of a maximum entropy distribution. In Section 3 we describe how the energetic cost associated with collisional friction is expressed as a constraint used in the maximization method. In Section 4 we show how the generalized Pareto distribution is obtained as a maximum entropy distribution. In Section 5 we describe the probabilistic properties and significance of the energetic cost. We

consider the implications of the analysis in the final section. In the fourth companion paper (Furbish et al., 2020c) we step back and examine the philosophical underpinning of the statistical mechanics framework for describing sediment particle motions and transport.

## 2 Background

### 2.1 Elements of the distribution of travel distances

With reference to Figure 1, let $x$ denote the particle travel distance with probability density function $f_x(x)$. The theoretical formulation (Furbish et al., 2020a) then begins with the particle disentrainment rate function defined by

$$P_x(x) = \frac{f_x(x)}{1 - F_x(x)} = \frac{f_x(x)}{R_x(x)} \, . \tag{1}$$

Here, $R_x(x) = 1 - F_x(x)$ is the exceedance probability function where $F_x(x)$ is the cumulative distribution function. The disentrainment rate $P_x(x)$ may be interpreted as a conditional probability per unit distance. Namely, upon multiplying both sides of Eq. (1) by $\mathrm{d}x$, then $P_x(x)\mathrm{d}x = f_x(x)\mathrm{d}x/R_x(x)$ is interpreted as the probability that a particle will become disentrained within the small interval $x$ to $x + \mathrm{d}x$, given that it "survived" travel to the distance $x$. In turn, upon rearranging Eq. (1) and





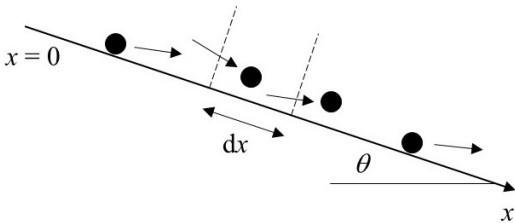

**Figure 1.** Definition diagram of surface inclined at angle $\theta$ and control volume with edge length $\mathrm{d}x$ through which particles move. Figure reproduced from companion paper (Furbish et al., 2020a).

making use of the fact that $f_x(x) = -\mathrm{d}R_x(x)/\mathrm{d}x$, the density $f_x(x)$ is obtained from

$$f_x(x) = P_x(x)e^{-\int_0^x P_x(x')\,\mathrm{d}x'}. \tag{2}$$

Thus, the significance of the disentrainment rate function becomes clear: it completely determines the density $f_x(x)$ via Eq. (2). For reference below, Eq. (1) and Eq. (2) are standard elements of survival (or reliability) analysis, without reference to entropy.

The particle energy balance formulated in Furbish et al. (2020a) leads to the result that for a given particle size and shape the disentrainment rate on an inclined surface with uniform slope and roughness is

$$P_x(x) = \frac{1}{Ax + B}. \tag{3}$$

Substituting Eq. (3) into Eq. (2) then leads to the generalized Pareto distribution,

$$f_x(x) = \frac{B^{1/A}}{(Ax + B)^{1+1/A}}. \tag{4}$$

where $A \in \Re$ is a shape parameter and $B > 0$ is a scale parameter (Pickands, 1975; Hosking and Wallis, 1987). The cumulative distribution is

$$F_x(x) = \begin{cases} 1 - \frac{B^{1/A}}{(Ax+B)^{1/A}} & A \neq 0 \\ 1 - e^{-x/B} & A = 0, \end{cases} \tag{5}$$

and the exceedance probability is

$$R_x(x) = \begin{cases} \frac{B^{1/A}}{(Ax+B)^{1/A}} & A \neq 0 \\ e^{-x/B} & A = 0. \end{cases} \tag{6}$$

For $A < 1$ the mean is

$$\mu_x = \frac{B}{1 - A}, \tag{7}$$





and for $A < 1/2$ the variance is

$$\sigma_x^2 = \frac{B^2}{(1-A)^2(1-2A)} \, . \tag{8}$$

The mean is undefined for $A \geq 1$ and the variance is undefined for $A \geq 1/2$.

In mechanical terms the shape and scale parameters $A$ and $B$ are

$$A = \frac{\alpha}{\gamma}\left[\frac{S}{\mu} - 1 + \frac{1}{\alpha}(\gamma - 1)\right] \qquad \text{and} \tag{9}$$

$$B = \frac{\alpha}{\gamma}\frac{E_{a0}}{mg\mu\cos\theta} \, . \tag{10}$$

Here, $S$ is the magnitude of the slope inclined at an angle $\theta$, $m$ is particle mass, $g$ is acceleration due to gravity, $\mu$ is a friction factor due to extraction of particle kinetic energy $E_p = (m/2)u^2$ where $u$ is the surface-parallel particle velocity, $E_a = \langle E_p \rangle$ is the arithmetic average particle energy so that $E_{a0}$ is the initial average energy at $x = 0$, $\gamma = E_a/E_h$ where $E_h$ is the harmonic average particle energy, and $\alpha = \alpha_0/(1 - \mu_1 Ki)$ where $\alpha_0$ and $\mu_1$ are factors of order unity and $Ki$ is the Kirkby number defined by

$$Ki = \frac{S}{\mu} \, , \tag{11}$$

which represents the ratio of gravitational heating to frictional cooling. Here we emphasize that $mg\cos\theta$ in Eq. (10) is *not* to be interpreted as the static normal weight of the particle, and $\mu$ is not interpreted as a Coulomb-like friction coefficient. Rather, $\mu \sim \langle \beta_x \rangle$, where $\langle \beta_x \rangle$ denotes the expected proportion of particle kinetic energy extracted per particle-surface collision during downslope motion. Details are provided in Furbish et al. (2020a, 2020b).

For plotting purposes we define a characteristic particle cooling distance $X = E_{a0}/mg\mu\cos\theta$ and in turn define the following dimensionless quantities denoted by circumflexes:

$$x = X\hat{x}, \quad E_a = E_{a0}\hat{E}_a \quad \text{and} \quad E_h = E_{a0}\hat{E}_h \, . \tag{12}$$

In addition, $a = A$ and $b = (\alpha/\gamma)\hat{E}_{a0}$. Then the dimensionless form of the generalized Pareto distribution, Eq. (4), is written as

$$f_{\hat{x}}(\hat{x}) = \frac{b^{1/a}}{(a\hat{x} + b)^{1+1/a}} \, , \tag{13}$$

For $a < 0$ the density $f_{\hat{x}}(\hat{x})$ is bounded at $\hat{x} = b/|a|$ (Figure 2). This density increases with $\hat{x}$ for $a < -1$, it is uniform for $a = -1$, and it decreases with $x$ for $a > -1$. It is triangular for $a = -1/2$. For $a = 0$ the density $f_{\hat{x}}(\hat{x})$ is exponential. For $a > 0$ this density is heavy-tailed. For $a \geq 1$ the mean of $f_{\hat{x}}(\hat{x})$ is undefined; and for $a \geq 1/2$ the variance is undefined.

We note that the definition of the differential entropy given in the next section involves the logarithm of the probability density function. In a strict sense this is acceptable only if the density is expressed in dimensionless form as in Eq. (13), or





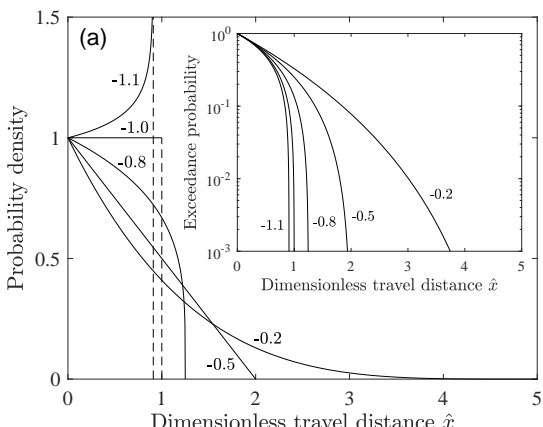 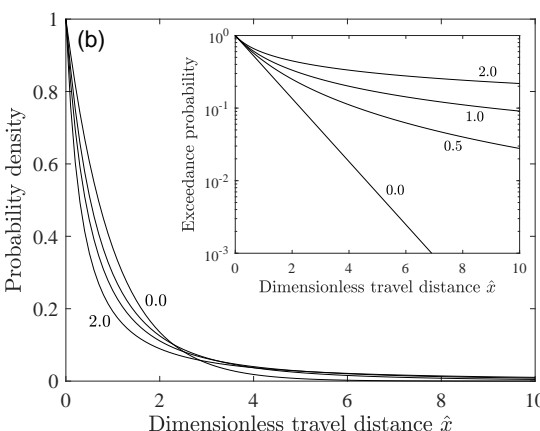

**Figure 2.** Plot of dimensionless probability density $f_{\hat{x}}(\hat{x})$ versus dimensionless travel distance $\hat{x}$ for scale parameter $b = 1$ and different values of the shape parameter $a$ for (**a**) $a < 0$ and (**b**) $a \geq 0$ with associated exceedance probability plots (insets). Figure reproduced from companion paper (Furbish et al., 2020a). Compare with Figure 1 in Hosking and Wallis (1987).

if the definition involves a discrete probability mass function. Nonetheless, the maximization method removes this logarithm

such that the outcome is dimensionally the same whether one starts with the dimensional form or the dimensionless form of the density. For simplicity we use the dimensional form, Eq. (4). In addition, for simplicity in plotting we set the scale parameter $B = 1$ in calculated functions containing this parameter, and in several plots we use dimensional abscissa values (e.g., distance $x$) without reference to units, noting that these have the same visual appearance as if plotted using dimensionless values.

Following Furbish et al. (2020b) we calculate the quantities

$$R_* = R_x^A \quad \text{and} \quad x_* = \frac{A}{B}x + 1.$$  (14)

Based on Eq. (6), values of the modified exceedance probability $R_*$ and the dimensionless travel distance $x_*$ should collapse to a straight line in a log-log plot with slope of -1 (Figure 3). The data in this figure, spanning more than three orders of magnitude of the dimensionless travel distance $x_*$, are compiled from Furbish et al. (2020b; Figure 16 therein). Values of $A$ and $B$ are estimated from laboratory measurements of particle travel distances reported by Gabet and Mendoza (2012) and Furbish et al.

(2020b), and from field-based measurements of travel distances reported by DiBiase et al. (2017) and Roth et al. (2020). This plot does not prove, but nonetheless supports, the idea that the generalized Pareto distribution correctly describes the energetics of the behavior of rarefied particle motions for a variety of slope and surface roughness conditions. The data fits for individual experiments with detailed explanation are presented in Furbish et al. (2020b).



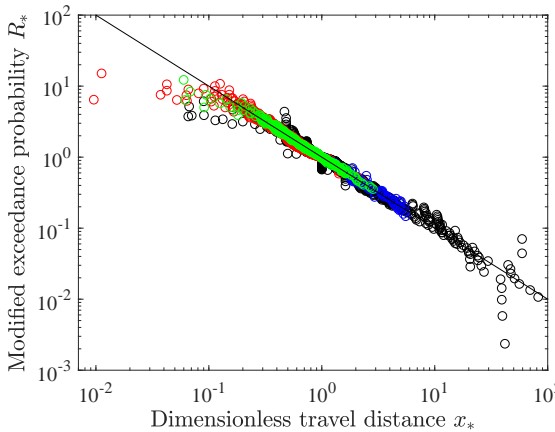

**Figure 3.** Plot of modified exceedance probability $R_*$ versus dimensionless travel distance $x_*$ and line with log-log slope of -1 for laboratory experiments described by Gabet and Mendoza (2012) (green) and Furbish et al. (2020b) (red) and field-based experiments described by DiBiase et al. (2017) (blue) and Roth et al. (2020) (black). Data for $A < 0$ fall to left of $x_* = 10^0 = 1$ with values in the tails represented by smaller values of $x_*$. Data for $A > 0$ fall to the right of $x_* = 10^0 = 1$ with values in the tails represented by larger values of $x_*$. Total data number is $N = 5671$.

## 2.2 Maximum entropy distribution

If $x$ denotes a continuous random variable with probability density $f_x(x)$ over $x = [0, \infty)$, then the differential entropy of $x$ is defined as

$$H(x) = -\int_0^\infty f_x(x) \ln f_x(x) \, \mathrm{d}x, \tag{15}$$

5 where it is understood that $f_x(x) \ln f_x(x) = 0$ when $f_x(x) = 0$. Given the lineage of this definition, hereafter we follow Peterson et al. (2013) and refer to it as the Boltzmann-Gibbs-Shannon (BGS) entropy. In turn, let $g_j(x)$ denote a measurable quantity of $x$ with $j = 0, 2, \ldots, n$. We then assume that

$$\mathrm{E}[g_j(x)] = \int_0^\infty g_j(x) f_x(x) \, \mathrm{d}x = a_j, \tag{16}$$

with finite $a_j$. For example, if $g_0(x) = g_0 = 1$, then Eq. (16) gives $a_0 = 1$. That is, the density $f_x(x)$ integrates to unity. If

10 $g_1(x) = x$, then Eq. (16) gives the mean of the distribution, $a_1 = \mu_x$. If $g_2(x) = (x - \mu_x)^2$, then Eq. (16) gives the variance, $a_2 = \sigma_x^2$. Note, however, that $g_j(x)$ need not be selected just to obtain the usual moments of a distribution. Indeed, Eq. (16) may represent a constraint imposed by a function $g_j(x)$ that does not coincide with a moment of $f_x(x)$. As described below, this is essential for heavy-tailed distributions whose first or second moment, or both of these moments, are undefined. The





maximum entropy distribution is then given by

$$f_x(x) = \exp\left[\sum_{j=0}^{n} \lambda_j g_j(x)\right],$$ (17)

where $\lambda_0, \lambda_1, \lambda_2, ...$ are Lagrange multipliers introduced in the problem of maximizing the entropy $H(x)$ (Appendix A). Moreover, as above we set $g_0(x) = g_0 = 1$ with $a_0 = 1$, which guarantees that the probability density $f_x(x)$ integrates to unity.

As a point of reference, a fixed mean with $g_1(x) = x$ and no other constraint leads to the result

$$f_x(x) = e^{\lambda_0} e^{\lambda_1 x}.$$ (18)

The Lagrange multipliers are then obtained as follows. By the definition of a probability density,

$$e^{\lambda_0} \int_0^\infty e^{\lambda_1 x}\,\mathrm{d}x = 1,$$ (19)

which leads to $e^{\lambda_0} = -\lambda_1$. Alternatively, Eq. (18) sometimes is presented as (e.g., Tolman, 1938; Schrödinger, 1946; Furbish and Schmeeckle, 2013)

$$f_x(x) = \frac{e^{\lambda_1 x}}{\int_0^\infty e^{\lambda_1 x}\,\mathrm{d}x},$$ (20)

where it becomes clear that $e^{\lambda_0}$ is a normalization factor that ensures the probability density integrates to unity. In turn, by the definition of the mean,

$$-\lambda_1 \int_0^\infty x e^{\lambda_1 x} = \mu_x,$$ (21)

which leads to $\lambda_1 = -1/\mu_x$ and the exponential distribution,

$$f_x(x) = \frac{1}{\mu_x} e^{-x/\mu_x},$$ (22)

where it becomes clear that the Lagrange multiplier $\lambda_1$ enforces the constraint of a fixed mean. The Gaussian distribution is similarly obtained as the maximum entropy distribution with the constraint imposed by a fixed second moment (variance).

The canonical example of the Boltzmann distribution of particle energy states is obtained in this manner as a maximum entropy distribution, where the mean is independently determined to be $k_B T$ (e.g., Schrödinger, 1946). The imposed constraints consist of extensive quantities that scale with system size: a fixed number of particles and a fixed total energy, which together guarantee a fixed mean energy. In a similar manner, Furbish and Schmeeckle (2013) and Furbish et al. (2016) derive an exponential distribution for the streamwise velocity states of particles transported as bed load, with the mechanical constraint imposed by a fixed total particle momentum under equilibrium transport conditions.

Our next task is to adapt these ideas to the generalized Pareto distribution, which is not among the exponential family of distributions. We note that there is a continuing effort given to this topic, notably in relation to heavy-tailed (non-exponential) distributions. Peterson et al. (2013) summarize the basis of these efforts, and note that one approach for inferring non-exponential





distributions is to appeal to nontraditional definitions of the entropy, for example, the Tsallis entropy (Tsallis, 1988), rather than the canonical BGS entropy. The procedure is the same: to maximize the defined entropy subject to an extensive constraint that scales with the system size. Here, however, we adopt the view of Peterson et al. (2013), who highlight the conclusions of

Shore and Johnson (1980). Namely, because the BGS definition of entropy uniquely ensures addition and multiplication rules of probability, any other definition of entropy yields a bias in the fitting of data. Peterson et al. (2013) suggest that this offers a "compelling first-principles basis for defining a proper variational principle for modeling distribution functions." Like these authors in their analysis of the energetics associated with the economics of scale, we retain the BGS definition of entropy and seek a non-extensive energy constraint aligned with the mechanics of the rarefied particle motion problem.

## 3   Energetic cost as a maximizing constraint

In the canonical example of the Boltzmann distribution, the particle energy state is an instantaneous quantity. Similarly, in the example of bed load particle velocities (Furbish and Schmeeckle, 2013; Furbish et al., 2016), the velocity state is an instantaneous quantity. The state of a particle changes from one instant to the next, and this state can be reached from smaller or larger state values. In these cases, the total particle energy and the total streamwise momentum are well-defined extensive

quantities such that the moments of the distributions are fixed. In the absence of additional information, the maximum entropy distribution must be among the exponential family.

In contrast to instantaneous quantities, the particle travel distance $x$ is an integrated quantity that reflects a dynamical particle history starting from the state $x = 0$. The state $x$ must be reached from smaller (unrecorded) state values; it cannot be reached from larger state values. Moreover, travel distances are not like an extensive quantity that scales linearly with the system size.

Nonetheless, particle motions require a source of energy and dissipation of energy. Following Peterson et al. (2013) we assume that the outcome of motions — the travel distances $x$ — can be represented in terms of an energetic cost that probabilistically constrains the organization of a great number of particles into accessible states $x$.

The disentrainment rate $P_x(x)$ has special significance in defining the energetic cost. In particular, this rate determines the energetic cost associated with reaching the state $x$. We start by using Eq. (10) to rewrite Eq. (3) as

$$P_x(x) = \frac{1}{B + Ax} = \frac{(\gamma/\alpha)mg\mu\cos\theta}{E_{a0}\left(1 + \frac{A}{B}x\right)}. \tag{23}$$

The denominator in Eq. (23) describes how the average particle energy $E_a(x)$ varies with $x$, whether this involves net cooling ($A < 0$), isothermal conditions ($A = 0$) or net heating ($A > 0$). The quantity $mg\mu\cos\theta$ in the numerator is the expected spatial rate at which energy is extracted by collisional friction, modulated by the factor $\gamma/\alpha$. Thus, the disentrainment rate represents the local relative energetic cost — the spatial rate at which particle energy is extracted per unit kinetic energy available during

motion at position $x$.





Earth **Surface**
**Dynamics**
Discussions
EGU

In turn, the relative energy extracted within a small interval $\mathrm{d}x$ is $P_x(x)\mathrm{d}x$ so the cumulative energy extracted per unit kinetic energy available is

$$w(x) = \int_0^x P_x(x')\,\mathrm{d}x'. \tag{24}$$

This is the cumulative energetic cost in reaching position $x$. For isothermal conditions ($A = 0$) the cumulative cost is

$$w(x) = \frac{1}{B}x. \tag{25}$$

For non-isothermal conditions ($A \neq 0$) the cumulative cost is

$$w(x) = \frac{1}{A}\ln\left(\frac{A}{B}x + 1\right). \tag{26}$$

These two expressions for $w(x)$ converge at small $x$ (Figure 4). Relative to the linear cumulative cost of isothermal conditions,

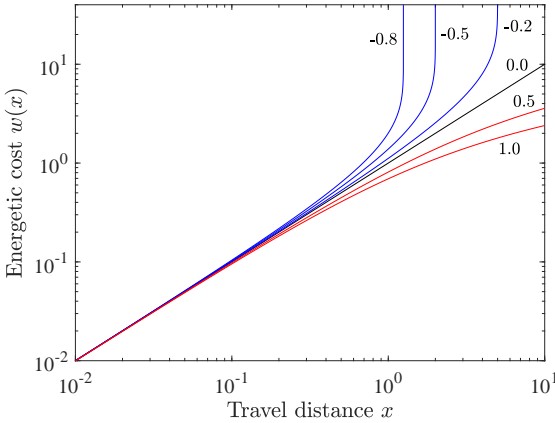

**Figure 4.** Plot of cumulative energetic cost $w(x)$ versus distance $x$ for several values of the shape parameter $A$ representing net cooling ($A < 0$, blue), isothermal conditions ($A = 0$, black) and net heating ($A > 0$, red).

Eq. (25), the cumulative cost with net cooling ($A < 0$) increases more rapidly up to the limiting distance given by $x = B/|A|$, and the cumulative cost with net heating ($A > 0$) increases more slowly with increasing distance $x$.

Consider first the isothermal case to illustrate the significance of the cost $w(x)$. This cost increases linearly with the distance $x$. Let $N$ denote a great number of particles. Among all accessible microstates — the many ways of arranging $N$ particles into states $x$ where each arrangement has a fixed total cost — most microstates involve particles with small state values and fewer with large state values. As shown below, this constraint leads to an exponential distribution. Note that Furbish and Schmeeckle (2013) provide a detailed description of the analysis leading to this outcome, including the basis for counting microstates (see Figure 3 and Appendix B therein), as applied to particle momentum states rather than travel distance states $x$. Nonetheless the analysis is otherwise conceptually identical. Tolman (1938) and Schrödinger (1946) provide clear descriptions of the canonical problem (in particular see Chapter II, "The Method of the Most Probable Distribution," in Schrödinger's text).





With non-isothermal conditions and net heating, it is easier to achieve larger state values than with isothermal conditions. Among all accessible microstates, an increasing proportion will have particles in larger states than would be predicted with a uniform cost rate. In contrast, with net cooling a smaller proportion of microstates will have particles in large states $x$ with an increasing relative cost to achieve these large states. Indeed, there is a limit on available energy to be spent in frictional cooling such that the relative cost goes to infinity at $x = B/|A|$. As shown below, these constraints lead to the generalized

Pareto distribution.

The energetic cost $w(x)$ is a natural choice for constraining the maximization method. As described in Section 6 (Discussion and conclusions), this choice is identical in form to the language of "cost" in the economics of scale (Peterson et al., 2013) leading to non-exponential (heavy-tailed) distributions of state values. We use these ideas next in deriving the maximum entropy distribution.

## 4  Generalized Pareto distribution

### 4.1  Constraints

Focusing on the generalized Pareto distribution, as above we start with the constraint given by $g_0(x) = g_0 = 1$, namely,

$$\mathrm{E}[g_0(x)] = \int_0^\infty f_x(x)\,\mathrm{d}x = 1\,. \tag{27}$$

A second, strong mechanical constraint is provided by assuming that the total cumulative energetic cost associated with colli-

sional friction is fixed. Starting with Eq. (24),

$$g_1(x) = \int_0^x P_x(x')\,\mathrm{d}x' = w(x)\,, \tag{28}$$

which is the cumulative energy extracted by friction per unit kinetic energy available in reaching position $x$. Then,

$$\mathrm{E}[g_1(x)] = \int_0^\infty \int_0^x P_x(x') f_x(x)\,\mathrm{d}x'\,\mathrm{d}x = \mu_w\,, \tag{29}$$

which is the average cumulative cost.

Starting with isothermal conditions ($A = 0$), the disentrainment rate $P_x(x) = P_x = 1/B$. This gives

$$\mathrm{E}[g_1(x)] = \frac{1}{B}\int_0^\infty x f_x(x)\,\mathrm{d}x = \mu_w = \frac{\mu_x}{B}\,, \tag{30}$$

which shows that the expected cumulative relative cost is unity with $\mu_x = B$. This is nominally the same as saying that the expected absolute cost is equal to the initial available energy $E_{a0}$. More generally with $P_x(x) = 1/(Ax + B)$,

$$\mathrm{E}[g_1(x)] = \frac{1}{A}\int_0^\infty \ln\left(\frac{A}{B}x + 1\right) f_x(x)\,\mathrm{d}x = \mu_w\,. \tag{31}$$

We use these two results in the maximization of entropy.





## 4.2 Maximization

For isothermal conditions, using Eq. (27) and Eq. (30) maximization leads to (Appendix A)

$$\ln f_x(x) - \lambda_0 - \lambda_1 \frac{1}{B} x = 0 \qquad \text{or} \tag{32}$$

$$f_x(x) = e^{\lambda_0} e^{\lambda_1 x/B}. \tag{33}$$

5 With $e^{\lambda_0} = -\lambda_1/B$ and $\lambda_1 = -1$ this becomes the exponential distribution,

$$f_x(x) = \frac{1}{B} e^{-x/B}, \tag{34}$$

with $B = \mu_x$.

More generally, using Eq. (27) and Eq. (31) maximization leads to (Appendix A)

$$\ln f_x(x) - \lambda_0 - \lambda_1 \frac{1}{A} \ln\left(\frac{A}{B} x + 1\right) = 0 \qquad \text{or} \tag{35}$$

$$f_x(x) = e^{\lambda_0} e^{(\lambda_1/A)\ln(Ax/B+1)} \tag{36}$$

$$= e^{\lambda_0} \left(\frac{A}{B} x + 1\right)^{\lambda_1/A}. \tag{36}$$

With $e^{\lambda_0} = -(A + \lambda_1)/B$ and $\lambda_1 = -(A+1)$ this becomes the generalized Pareto distribution given by Eq. (4), thus showing
15 that this distribution is a maximum entropy distribution.

## 5 Properties of the energetic cost

### 5.1 Cumulative energetic cost

Because of the importance of the energetic cost as a constraint in the maximum entropy method, here we examine the properties
of this cost. The cumulative energetic cost $w$ is a monotonic function of the travel distance $x$ according to Eq. (25) and Eq.
20 (26), so we can readily deduce (Appendix B) the probability density function $f_w(w)$ of the cost $w$. For isothermal conditions
$(A = 0)$ this density is

$$f_w(w) = e^{-w}, \tag{37}$$

with mean $\mu_w = 1$. The cumulative distribution is

$$F_w(w) = 1 - e^{-w}. \tag{38}$$




For non-isothermal conditions ($A \neq 0$) the density is

$$f_w(w) = e^{1/A} e^{Aw} e^{-(1/A)e^{Aw}},$$ (39)

which has attributes of an extreme value distribution. The mean is

$$\mu_w = -\frac{e^{1/A}}{A} \text{Ei}\left(-\frac{1}{A}\right),$$ (40)

where Ei denotes the exponential integral. The cumulative distribution is

$$F_w(w) = 1 - e^{1/A} e^{-(1/A)e^{Aw}}.$$ (41)

Note that these functions depend on the shape parameter $A$ but not on the scale parameter $B$.

Whereas the generalized Pareto distribution of travel distances $x$ for net cooling ($A < 0$) is bounded at $x = B/|A|$ (Figure 2), the probability density $f_w(w)$ of energetic costs $w$ is unbounded (Figure 5). For isothermal conditions ($A = 0$) the cost $w$

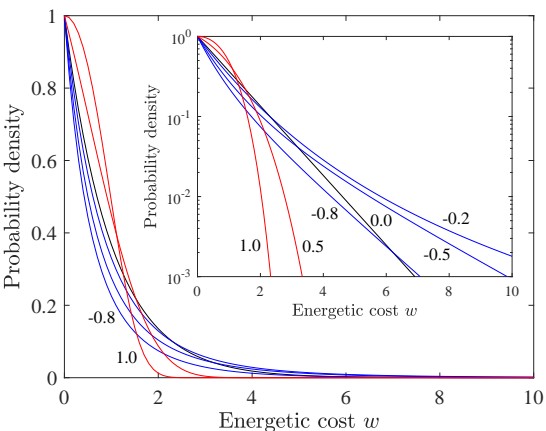

**Figure 5.** Plot of probability density $f_w(w)$ of energetic cost $w$ for different values of the shape parameter $A$, with semi-log plot (inset) showing heavy-tailed form ($A < 0$, blue) and light-tailed form ($A > 0$, red).

is linearly related to the travel distance $x$ so the distribution $f_w(w)$ has the same exponential form as $f_x(x)$. With net cooling ($A < 0$) the distribution $f_w(w)$ is heavy-tailed and with neat heating ($A > 0$) it is light-tailed. With cooling the energetic cost $w$ increases with distance $x$ up to $x = B/|A|$ so probability is shifted to larger values of $w$. With heating the energetic cost decreases with distance $x$ so probability is shifted to lower values of $w$ with increasing $A$. Over the domain $-1 \leq A \leq 1$ the average cost $\mu_w$ has a maximum at an intermediate value of $A \approx -0.33$ (Figure 6). For conditions to the left of the maximum

the relative costs of motions are large but the travel distances are small. For conditions to the right of the maximum the travel distances are larger but with smaller relative costs. For conditions of net heating ($A > 0$) the travel distances increase but the relative costs decrease.





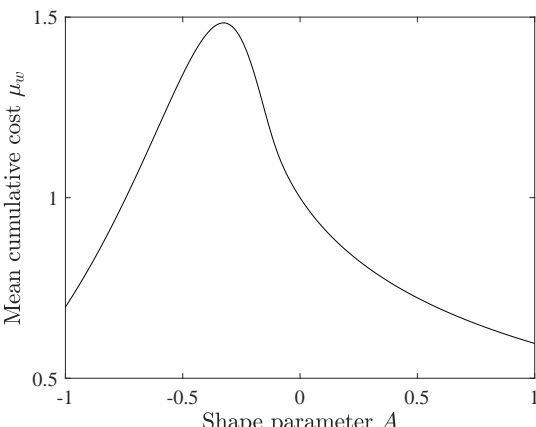

**Figure 6.** Plot of mean energetic cost $\mu_w$ versus shape parameter $A$, showing maximum value at $A \approx -0.33$.

The total cumulative cost $W(w)$ up to the value $w$ is

$$W(w) = \int_0^w w' f_w(w') \, \mathrm{d}w'. \tag{42}$$

5  Alternatively, the total cumulative cost up to the distance $x$ is

$$W(x) = \int_0^x w(x') f_x(x') \, \mathrm{d}x'. \tag{43}$$

Expressions for Eq. (42) and Eq. (43) are provided in Appendix B and show how the total costs $W(w)$ and $W(x)$ grow with increasing $w$ and $x$ to a finite value. Consider here the product $W_*(x) = w(x) f_x(x) = \mathrm{d}W(x)/\mathrm{d}x$, which is the total cost per unit travel distance. This function is like a frequency-magnitude product and reflects the relative contribution to the total cost of different parts of the travel distance domain (Figure 7). For net cooling ($A < 0$) and large negative $A$ the total cost is dominated by the high individual costs of the largest travel distances near the upper bound given by $x = B/|A|$. With increasing $A$ the cost becomes more evenly distributed. At isothermal conditions ($A = 0$) the total cost is dominated by travel distances near the mean distance. For net heating the total cost is dominated by the relatively large individual costs and proportions of small travel distances, although the contribution of large travel distances grows with increasing $A$.

## 5.2 Frictional loss to heat

The energetic cost outlined above pertains to the conversion of translational kinetic energy into other forms, including rotational energy, surface deformation and heat — all under the heading of collisional friction. This cost, however, is not the same as the total energy conversion to heat.

Consider the total energy extracted by friction and ultimately converted to heat. Note first that the quantity $mg \sin \theta$ at first glance normally is interpreted as the downslope component of the weight of a particle (or control volume) with mass $m$. In

Earth **Surface**
**Dynamics**
Discussions

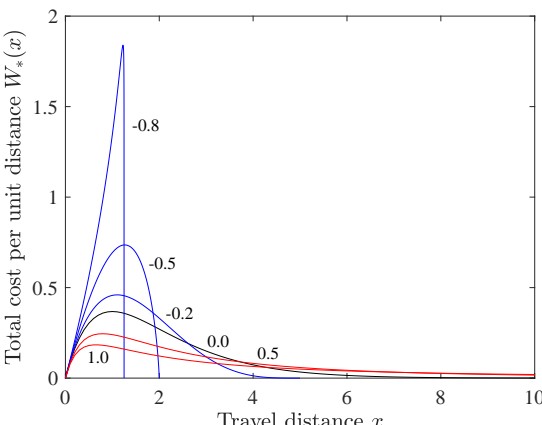

**Figure 7.** Plot of the total cost per unit travel distance $W_*(x)$ versus travel distance $x$ for different values of the shape parameter $A$ representing net cooling ($A < 0$, blue), isothermal conditions ($A = 0$, black) and net heating ($A > 0$, red).

energetic terms, however, this quantity is to be interpreted as the accessible gravitational potential energy per unit downslope travel distance (Furbish et al., 2020a). For an individual particle traveling a distance $x$ the heat generated is

$$q_p(x) = E_{p0} + mg\sin\theta x. \tag{44}$$

Taking the ensemble average of Eq. (44) and using Eq. (7),

$$\mu_{q_p} = E_{a0} + mg\sin\theta \frac{B}{1-A} \qquad A < 1. \tag{45}$$

The total heat generated by $N$ particles is then $N\mu_{q_p}$. As a fun point of reference, 100 particles, each with a diameter of 0.1 m and an average starting velocity of 1 m s$^{-1}$ traveling an average distance of 10 m down a 30 degree slope, produce about 0.32 J of heat — the equivalent of an ordinary 100 W light bulb turned on for 0.0032 s. On the other hand, for a million similar
particles traveling an average of 100 m down a 45 degree slope, we must leave the light bulb on for nearly eight minutes.

This result offers an example of how application of the maximum entropy method can be misleading. Namely, suppose we assume that a total fixed quantity of heat generated by particle motions, because this is an energetic "cost," provides a constraint on the maximization procedure. In this situation, and with no further constraints, the maximum entropy method leads to an exponential distribution $f_{q_p}(q_p)$ of heat states $q_p$ with mean $\mu_{q_p} = E_{a0} + mg\sin\theta\mu_x$. Because $q_p$ and $x$ are linearly
related, then using Eq. (B1) (Appendix B) the distribution $f_x(x)$ of travel distances $x$ would be exponential. Note that at this point, however, the mean travel distance $\mu_x$ is not well constrained, as no mechanical information is provided for how particles achieve the distance states $x$. Whereas the choice of an exponential distribution for $f_{q_p}(q_p)$ is a maximally unbiased choice, it almost certainly is incorrect. We comment further on this type of naïve use of the maximum entropy method below.





## 6 Discussion and conclusions

Let us acknowledge that a distribution identified as a maximum entropy distribution based on empirically constraining one or more of its moments is not necessarily a special outcome. For example, we frequently fit data to exponential and Gaussian distributions based on estimates of the mean and variance of these distributions — assuming these moments exist and are finite — without reference to maximum entropy. In other words, asserting that a random variable possesses a finite expected value (mean or variance) and then using this assertion to choose the distribution based on the maximum entropy method has no meaningful mechanical significance if the mechanical basis of the constraint is not specified. In this situation a maximum entropy criterion is just one among numerous inferential methods — albeit with the decided merit of being maximally indifferent in the choosing of the distribution. Only when the constraining moment has independent mechanical meaning, and in the absence of additional information, does the label of maximum entropy carry mechanical significance. The example of heat states $q_p$ described in Section 5.2 illustrates this point.

For example, Furbish et al. (2016) suggest:

"In focusing on the mechanical side of the duality of Jaynes's principle [of maximum entropy], it becomes important to distinguish between a "strong" mechanical constraint, a "weak" mechanical constraint, and an empirical constraint, as these inform confidence in the resulting choice of a distribution... A strong mechanical constraint is one that derives directly from a dynamics argument... A weak constraint is one that derives from a mechanical definition, for example, an appeal to mass conservation... An empirical constraint is one that appeals to our confidence in suggesting a general behavior from experiments or dimensional analysis but lacks a clear dynamics underpinning."

For rarefied bed load particles transported under equilibrium conditions, Furbish et al. (2016) show that the condition of fixed total particle momentum provides a strong mechanical constraint. In this situation the maximum entropy method predicts an exponential distribution of particle velocities in the absence of any additional mechanical information — consistent with measurements of particle velocities based on high-speed imaging (e.g., Lajeunesse et al., 2010; Roseberry et al., 2012; Furbish and Schmeeckle, 2013; Fathel et al., 2015; Wei et al. 2015). We suggest that the total cumulative energetic cost used herein to constrain the maximum entropy method similarly represents a strong mechanical constraint.

As a point of reference, the analysis presented herein is akin to the energetics associated with the economics of scale as examined by Peterson et al. (2013). To illustrate this idea we start with a binomial expansion of the disentrainment rate, Eq. (3), to give

$$P_x(x) = \frac{1}{B}\left(1 - \frac{A}{B}x + ...\right). \tag{46}$$

Momentarily focusing on the leading and first-order terms for illustration, Eq. (46) has the same form as the "communal cost-minus-benefit function" proposed by Peterson et al. (2013, Eg. (5) therein; Appendix C). Using the language of economic costs, here the state $x$ may be interpreted as the size of a community, for example, "particles forming colloidal clusters, or social processes such as people joining cities, citations added to papers, or link creation in a social network" (Peterson et al.,



2013, p. 20381). The leading term in Eq. (46) may be interpreted as an intrinsic cost for an individual to achieve ("join") the state $x$. For $A > 0$ the first-order term represents a "discount" provided by the community of size $x$. For $A < 0$ the first-order term represents a "penalty" imposed by the community. If the cost is independent of size ($A = 0$), then the cost rate is

fixed ($P_x = 1/B$) and the maximum entropy method leads to an exponential distribution of states $x$. If the cost is shared with increasing size ($A > 0$), then the cost of joining the state $x$ decreases with increasing size. This means that larger sizes (states) are more likely to occur than if a discount is not provided, leading to a heavy-tailed distribution of states $x$. Conversely, if joining a state $x$ involves a penalty ($A < 0$), then exclusion with increasing size occurs, leading to a light-tailed or bounded distribution of states $x$. In this analysis the idea of cost is fundamentally energetic, whether involving free energy for colloid

particles, or the energy consumed by individuals in joining some form of social construct.

When rearranged, the "cost-minus-benefit" function proposed by Peterson et al. (2013) yields a cost function (Appendix C) whose form is identical to that of the disentrainment rate, Eq. (3). In the economics of scale problem the costs are nominally absolute energetic costs. In the problem of rarefied particle motions the cost function (i.e., the disentrainment rate) represents the local relative energetic cost. Nonetheless, the formalism involving a fixed total cumulative cost is essentially the same. With

net particle heating it becomes easier for particles to achieve larger states $x$ relative to a fixed local energetic cost, analogous to effects of a discount in the economics of scale. With net cooling this effect is reversed, where the local relative energetic cost increases with the state $x$. The key mathematical construct of the disentrainment rate, Eq. (3), is that the state $x$ appears in the denominator of this cost function.

In this problem the maximum entropy method in effect considers all possible accessible microstates — the many different

ways to arrange a great number of particles into distance states $x$ where each arrangement satisfies the same fixed total energetic cost. (Figure 3 in Furbish and Schmeeckle (2013) illustrates this idea.) Then, the generalized Pareto distribution uniquely represents the most probable arrangement. This idea equally applies to the accessible microstates associated with net cooling, isothermal conditions and net heating. To elaborate this point, consider the upper bound on travel distances, $x = B/|A|$, under conditions of net cooling. This is the distance at which, according to Eq. (23), the expected available kinetic energy goes to

zero such that the disentrainment rate $P_x(x)$ becomes unbounded. From this perspective, the conditional probability $P_x(x)\mathrm{d}x$ that motions cease within a small interval $\mathrm{d}x$ approaches unity as $x \to B/|A|$. However, this is not to be interpreted as a "hard" boundary determined by mechanical behavior. Rather, according to Eq. (26) the cumulative energetic cost becomes unbounded at the distance $x \to B/|A|$. For a small upper bound $x = B/|A|$ the total energetic cost involves contributions from all particle motions but is dominated by the large individual costs of the largest travel distances near this upper bound (Figure 7). From

this perspective, the bounded form of the distribution is just the most probable among all possible arrangements satisfying the constraint of a fixed total cost, in this case dominated by the individual costs of the largest travel distances. A similar conclusion pertain to the bounded form of the distribution as contributions of individual motions to the total cost become more broadly distributed with increasing $A$ (Figure 7). In turn, no matter how large the upper bound $x = B/|A|$ becomes as $|A|$ approaches zero, this upper bound nonetheless remains finite. The generalized Pareto distribution then "flips" to an exponential form with unbounded distance states only in the limit of $A \to 0^-$. In approaching this limit, the basic physics of particle motions does not change. Similarly, in approaching this limit $A \to 0^+$ from the heavy-tailed form of the generalized





Pareto distribution, no changes in physics occur. That is, the essence of the balance between gravitational heating and frictional cooling by particle-surface collisions remains the same; there is nothing special or unusual about particle-surface interactions associated with crossing the isothermal transition. Thus, the most probable arrangement of distance states $x$ is in each case — net cooling, isothermal conditions and net heating — a reflection of the unifying probabilistic outcome associated with a fixed total energetic cost.

Here we return to Eq. (2), the standard formulation of the probability density $f_x(x)$ presented in survival analysis,

$$f_x(x) = P_x(x)e^{-\int_0^x P_x(x')\,\mathrm{d}x'},\tag{47}$$

and compare this with the entropy maximization criterion given by

$$f_x(x) = e^{\lambda_0}e^{\lambda_1\int_0^x P_x(x')\,\mathrm{d}x'}.\tag{48}$$

Assuming the Lagrange multiplier $\lambda_1 = -(A+1)$, then Eq. (48) becomes

$$f_x(x) = e^{\lambda_0}e^{-A\int_0^x P_x(x')\,\mathrm{d}x'}e^{-\int_0^x P_x(x')\,\mathrm{d}x'},\tag{49}$$

which has the form of Eq. (47) with

$$P_x(x) = e^{\lambda_0}e^{-A\int_0^x P_x(x')\,\mathrm{d}x'}.\tag{50}$$

Substituting $e^{\lambda_0} = 1/B$ and $P_x(x) = 1/(Ax+B)$ into Eq. (49) and evaluating the integrals confirms that the generalized Pareto distribution is retrieved.

We now have the interesting result that, for this problem, determining the distribution $f_x(x)$ according to Eq. (47) is the same as obtaining this distribution using a maximum entropy criterion. This occurs because the disentrainment rate $P_x(x)$ represents an energetic cost to particles reaching states $x$. Then, inasmuch as the total energetic cost probabilistically constrains the organization of a great number of particles into accessible states consistent with the maximization method, the resulting distribution must be a maximum entropy distribution. If instead the disentrainment rate function $P_x(x)$ is heuristically proposed or empirically fitted to data without reference to constraints imposed on the system, then the distribution obtained from Eq. (47) will be consistent with the disentrainment rate function, but this does not guarantee that the distribution is a maximum entropy choice.

The analysis presented here represents an unusual situation. Namely, the generalized Pareto distribution of travel distances and its parametric values are known a priori, and this distribution is then shown to be a maximum entropy distribution consistent with the constraint imposed by a fixed energetic cost. In contrast, normally the distribution is not known and the maximum entropy method is used to choose the distribution in an unbiased manner based on known constraints — as exemplified by the Boltzmann distribution. As emphasized by many, starting with Jaynes (1957a), the maximum entropy method represents a compelling strategy for choosing a distribution. Nonetheless, it is important to highlight the fact that a distribution thus chosen is not necessarily the "correct" distribution (Furbish et al., 2016). Rather, a distribution derived from a maximum entropy criterion is unbiased in that it is faithful to what is known mechanically, but no more; it is the most reasonable choice in the





absence of additional information. In this sense the maximum entropy method is a formal application of Occam's razor. Thus, the value of showing that the generalized Pareto distribution is a maximum entropy distribution is this: the analysis represents a novel generalization of an energy-based constraint in using the maximum entropy method to infer non-exponential distributions

— to include the versatile properties (forms) of the generalized Pareto distribution as applied to the rarefied particle motion problem. Importantly, the analysis uses the BGS definition of entropy rather than a nontraditional definition. We suggest that this result offers promise for examining particle motions in other systems, including particles transported as bed load, where insights involving particle energetics might become useful as we learn more about the physics involved.

*Data availability.* The data plotted in Figure 3 are available from sources described in Furbish et al. (2020b).

**Appendix A: Maximization**

The maximization method involves the calculus of variations (Cover and Thomas, 1991), of which a version closer to the original analysis of Boltzmann is presented in Furbish and Schmeeckle (2013) and Furbish et al. (2016). Using the BGS definition of entropy given by Eq. (15) together with the constraints $g_0(x) = g_0 = 1$ and $g_1(x)$ given by Eq. (29) we form the following objective function:

$$J[f_x(x)] = \int_0^\infty f_x(x) \ln f_x(x) \, dx - \lambda_0^* \left[ \int_0^\infty f_x(x) \, dx - 1 \right]$$

$$-\lambda_1 \left[ \int_0^\infty \int_0^x P_x(x') f_x(x) \, dx' \, dx - \mu_w \right], \tag{A1}$$

with Lagrange multipliers $\lambda_0^*$ and $\lambda_1$. Taking the functional derivative of Eq. (A1) with respect to $f_x(x)$ and setting the result to zero then leads to

$$\ln f_x(x) - \lambda_0 - \lambda_1 \int_0^x P_x(x') \, dx' = 0, \tag{A2}$$

with $\lambda_0 = \lambda_0^* - 1$. This yields

$$f_x(x) = e^{\lambda_0} e^{\lambda_1 \int_0^x P_x(x') \, dx'} . \tag{A3}$$

For isothermal conditions with $P_x(x) = P_x = 1/B$, Eq. (A3) becomes

$$f_x(x) = e^{\lambda_0} e^{\lambda_1 x / B} . \tag{A4}$$

With $\mu_x = B$, evaluating the Lagrange multipliers gives $e^{\lambda_0} = -\lambda_1/B$ and $\lambda_1 = -1$ leading to Eq. (34) in the text. For non-isothermal conditions with $P_x(x) = 1/(Ax + B)$, Eq. (A3) becomes

$$f_x(x) = e^{\lambda_0} \left( \frac{A}{B} x + 1 \right)^{\lambda_1 / A} . \tag{A5}$$





With $\mu_x = B/(1-A)$, evaluating the Lagrange multipliers using l'Hôpital's rule gives $e^{\lambda_0} = -(A+\lambda_1)/B$ and $\lambda_1 = -(A+1)$ leading to Eq. (4) in the text.

### Appendix B: Total cumulative cost

Let $x$ denote a random variable with probability density $f_x(x)$. If a random variable $w$ is a monotonic function of $x$, namely $w = g(x)$, then the probability density $f_w(w)$ of $w$ is given by

$$f_w(w) = \left| \frac{\mathrm{d}}{\mathrm{d}w} \left[ g^{-1}(w) \right] \right| f_x[g^{-1}(w)]. \tag{B1}$$

For isothermal conditions ($A = 0$) the cumulative cost $w(x)$ is,

$$w(x) = \frac{1}{B} x, \tag{B2}$$

so $g^{-1}(w) = x = Bw$. Then $\mathrm{d}g^{-1}(x)/\mathrm{d}w = B$ and the probability density is

$$f_w(w) = B \frac{1}{B} e^{-Bw/B} = e^{-w}. \tag{B3}$$

The cumulative distribution is

$$F_w(w) = 1 - e^{-w}. \tag{B4}$$

The mean of this distribution is

$$\mu_w = \int_0^\infty w e^{-w} \, \mathrm{d}w = 1. \tag{B5}$$

For non-isothermal conditions ($A \neq 0$),

$$w(x) = \frac{1}{A} \ln \left( \frac{A}{B} x + 1 \right), \tag{B6}$$

so $g^{-1} = x = (B/A)(e^{Aw} - 1)$. Then $\mathrm{d}g^{-1}(x)/\mathrm{d}w = Be^{Aw}$ and

$$f_w(w) = Be^{Aw} \frac{1}{B} e^{-(B/A)(e^{Aw}-1)/B}$$

$$= e^{1/A} e^{Aw} e^{-(1/A)e^{Aw}}. \tag{B7}$$

The cumulative distribution is

$$F_w(w) = 1 - e^{1/A} e^{-(1/A)e^{Aw}}. \tag{B8}$$





Noting that for $A < 0$ the limit of Eq. (B6) as $x \to -B/A$ is $w \to \infty$ and for $A > 0$ the limit as $x \to \infty$ is $w \to \infty$, then the mean of the distribution is

$$\mu_w = e^{1/A} \int_0^\infty w e^{Aw} e^{-(1/A)e^{Aw}} \, \mathrm{d}w$$

$$= -\frac{e^{1/A}}{A} \mathrm{Ei}\left(-\frac{1}{A}\right), \tag{B9}$$

where Ei denotes the exponential integral.

The total cumulative cost $W(w)$ up to the value $w$ is

$$W(w) = \int_0^w w' f_w(w') \, \mathrm{d}w'. \tag{B10}$$

For isothermal conditions,

$$W(w) = \int_0^w w' e^{-w'} \, \mathrm{d}w' = 1 - (1+w)e^{-w}. \tag{B11}$$

For non-isothermal conditions,

$$W(w) = e^{1/A} \int_0^w w' e^{Aw'} e^{-(1/A)e^{Aw'}} \, \mathrm{d}w'$$

$$= \frac{e^{1/A}}{A} \left[ \mathrm{Ei}\left(-\frac{1}{A}e^{Aw}\right) - Awe^{-(1/A)e^{Aw}} \right]$$

$$-\frac{e^{1/A}}{A} \mathrm{Ei}\left(-\frac{1}{A}\right). \tag{B12}$$

The total cumulative cost $W(x)$ up to the distance $x$ is

$$W(x) = \int_0^x w(x') f_x(x') \, \mathrm{d}x'. \tag{B13}$$

For isothermal conditions,

$$W(x) = \frac{1}{B} \int_0^x \frac{1}{B} x' e^{-x'/B} \, \mathrm{d}x'$$

$$= 1 - (1 + x/B)e^{-x/B}. \tag{B14}$$



For non-isothermal conditions,

$$W(x) = \frac{1}{A} \int\limits_0^x \ln\left(\frac{A}{B}x' + 1\right) \frac{B^{1/A}}{(Ax' + B)^{1+1/A}} \, \mathrm{d}x'$$

$$= 1 - \left[\frac{1}{A}\ln\left(\frac{A}{B}x + 1\right) + 1\right] \frac{B^{1/A}}{(Ax + B)^{1/A}}. \tag{B15}$$

The total cumulative cost $W(x)$ systematically increases with increasing travel distance $x$ (Figure B1).

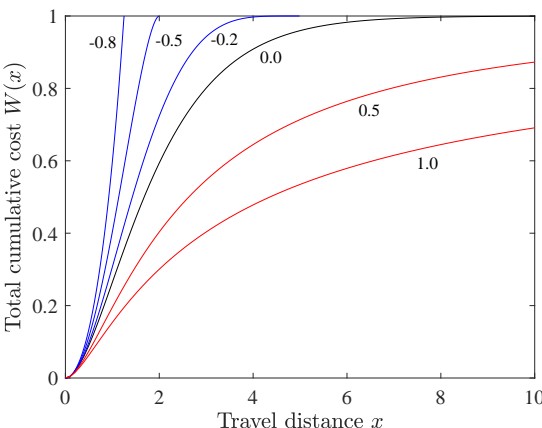

**Figure B1.** Plot of total cumulative cost $W(x)$ versus travel distance $x$ for different values of the shape parameter $A$ representing net cooling ($A < 0$, blue), isothermal conditions ($A = 0$, black) and net heating ($A > 0$, red).

## 5   Appendix C: Cost-minus-benefit function of Peterson et al. (2013)

Peterson et al. (2013) focus on discrete systems where the state $x \to k = 1, 2, 3, \ldots$ denotes a community size. Their cost-minus-benefit function has the form

$$\alpha_k = \alpha_0 - \frac{\alpha_k}{k_0}k. \tag{C1}$$

Here,

10     "the quantity on the left side of Eq. (C1) is the total cost-minus-benefit when a particle joins a $k$-mer community.
       The joining cost has two components, expressed on the right side: each joining event has an intrinsic cost $\alpha_0$ that
       must be paid, and each joining event involves some discount that is provided by the community. Because there
       are $k$ members of the existing community, the quantity $\alpha_k/k_0$ is the discount given to a joiner by each existing
       community particle, where $k_0$ is a problem-specific parameter that characterizes how much of the joining cost
       burden is shouldered by each member of the community."



Rearranging Eq. (C1) then gives

$$\alpha_k = \frac{\alpha_0 k_0}{k_0 + k}, \tag{C2}$$

which is analogous to the disentrainment rate function $P_x(x)$ given by Eq. (3). The key similarity between Eq. (C1) and Eq.
(3) is that $k$ and $x$ are in the denominators of these cost functions.

*Author contributions.* DJF wrote much of the paper with contributions by SGWW and THD.

*Competing interests.* We have no competing interests.

*Acknowledgements.* We acknowledge support by the U.S. National Science Foundation (EAR-1420831 and EAR-1735992). We appreciate continuing discussions with Peter Haff regarding entropy in Earth-surface systems. Nakul Deshpande offered useful reactions to an earlier draft.





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
