# Peer review of "Rarefied particle motions on hillslopes: 3. Entropy"

_Earth Surface Dynamics, 2020_

## Referee Comment (RC1) · Joris Heyman (Referee) · 1 Feb 2021

The third paper has been the hardest for me to follow, because it touches concepts from statistical physics, that are less frequent in the earth science research community. For what I understood, the author claim to generalize the maximum entropy principle to several energy-based physical constraints. With this approach, they find similar Pareto distributions as with the varying deposition probability framework developed in companion paper 1. I believe this is an important result that goes far beyond particle motion on hillslopes, so that I am not convinced that associating this study as a companion paper is a judicious choice. In my opinion, proposing this study to a more physically sound readership journal than esurf would have a greater impact (Physical review ?). However, I rely on the editor's and other reviewers point on view for this.

[Figure]

Other comments: p1 l15 ". . . that is heavy-tailed for net cooling and light tailed for net heating" Isn't it the other way around ?! (3) precise that A can be between -B and infinity ? (16) What is notation E[] for ? You have already used it for energy. . . p19 l 7: What is Occam's razor ?

Please also note the supplement to this comment:
https://esurf.copernicus.org/preprints/esurf-2020-100/esurf-2020-100-RC1-supplement.pdf

**ESurfD**

**Supplement:**

**Review of "Rarefied Particle motions on hillslopes" (Joris Heyman)**

**Global comments:**

These 4th companion papers are all very relevant for the different messages and new results they convey. I have fully enjoyed this tough but inspiring reading. I have no major comments to make, although, as explained in part 3 and 4, I suggest submitting the last two studies separately (in esurf or other journal), since their scope is much more general than the hill-slope problem.

While pleasant, the writing style contains many "didactic sidebars", "anecdotes" or humor that do not ease the understanding of an already complex message. I sometimes felt more like reading a book than a journal article (the 4th papers format do not help concision neither). Beside precise structural points (see part 1 of the review), I would tend to think that it is possible to globally shorten the text, summarizing the ides, without altering the important results and transferring extra materials in supplementary material.

**Specific comments:**

**1) Theory**

This first companion paper is the master piece of the serie, presenting all theoretical developments.

**State of the art** The literature review has been placed after the theoretical developments (Section 5 Related formulation), which, in my opinion, do not help to globally envision the originality of the proposed formulation with respect to existing ones, and understand the main challenges of the hillslope problem. I would suggest the authors to better highlight the originality of their approach based on a succinct literature review from the very beginning. This could also help to introduce the important variables.

**Summary of findings** In addition to this originality statement, I believe that a simple summary of findings should precede the detailed theoretical developments. In contrast to the book format, we expect in a journal article to have a rapid understanding of the main results. I had to wait for the summary provided in the second companion paper to make me a clear mental image of the main ingredients of the theory proposed, which I have expressed this way :

1. Particle Mass conservation dN/dx = - N/Ea

2. The variation of the ensemble average energy is constant (since forces are constant ?):  $dEa/dx = Cst \rightarrow Ea = Ax+B$

Thus, the mean disentrainment rate is P=-1/N dN/dx = 1/(Ax+B), and the PDF of travel distances is a Pareto distribution, in place of the classical Exponential distribution found when P is a constant. Such ultra-simplified preamble would ease a lot the navigation into the details of the theory latter on.

**Terminology** I understand the analogy between statistical physics of gas and motion of particles down a slope, although I am a bit skeptical on translating all the technical vocabulary for this situation. For instance, the terms *"thermal collapse"*, *"iso-thermal"* and *"net heating"* are not fully transparent with respect to gravity driven motions, and will remain obscure for a majority of readers. In my opinion the notion of *"heat"* in a gas refers to zero-mean velocity fluctuations, and is thus not perfectly suited to describe a net shift of mean velocities as is the case in non-equilibrium particle motion driven by gravity. I understand the authors conceive the thermal collapse as a net decrease of particle energy and the heating as a net increase of particle energy. However, if they would extend their statistical formulation to the evolution of higher statistical moments of energy states, there will be a confusion

between drift (mean velocity) and diffusion (fluctuations around the mean). My suggestion would be to simply use the transparent terms of mean "deceleration" and "acceleration" of particles ? One of the drawback of using energy balance instead of mass and momentum conservation is that well defined (and measurable) variables such as particle velocity and acceleration are lumped into an energy state, which is less tangible to the observer. Then, it is very easy to understand the disentrainment rate in terms of a decelerating particle (disentrainment probability growing with x, A>0) or accelerating motion (disentrainment probability decreasing with x, A<0).

**Fokker-Planck equation** I understand the authors objective to cast their analysis into a fully probabilistic framework, although I did not get the necessity here to derive a complete Fokker-Planck equation for E if none of the higher moment are used latter on. Indeed, the authors introduce beta2 (diffusivity of the energy state), which is never used afterwards. Why ? In my opinion, the shape of the pdf (Pareto) is only dependent upon the evolution of the disentrainment rate probability, not on the FP description of energy states. This is a 'simple' non-homogeneous Poisson process. Introducing the FP formulation is thus somewhat confusing for the main message. If this FP equation had an importance for the description of the difference between harmonic or algebraic average of the energy states (Ea, Eh), it might have been preferable to introduce this concept differently (I personally did not get this distinction entirely).

**Meta-stability :** Being familiar to the study of Quartier et al. 2000, I wondered if the theoretical description proposed by the authors is also able to explain the occurrence of meta-stable states of motion due to micro- roughness. Indeed, depending on the initial particle velocity, a particle may be trapped by bed roughness or continue its motion indefinitely. I would have liked to find a mention of this somewhere in the text.

Quartier, L., et al. "Dynamics of a grain on a sandpile model." *Physical Review E* 62.6 (2000): 8299.

Specific Points :

- p5 l8 : I did not get in which sense these probabilistic formulation are "scale independent"
- p5 l17 : "can be a **constant** determined"
- p8 l9 : "The law of the unconscious statistician" ...which means for an unconscious reader ?
- p9 l15: This sidebar could come before, at the beginning of the section
- p10 l20 : "So bear with us" . This do not presage good...
- p11 l25 : Think of moving this didactic sidebar in annexe
- p 12 l 23 : What does "immaterial" mean in this context ?
- -p15 : The authors mention "deposition" in granular gases. I do not understand well how particles can deposit in absence of boundary. Do the authors mean "aggregation" ?

-p16 (39) and (40) : beta and beta2 have the same units ?

-p19 l6 "disentrainment rate, consistent with the deposition rate." I do not understand this.

-p20 l25-30 This paragraph is very confusing for me. Could you reformulate it in simpler way ?

-p28 l17 **m** g mu cos theta

- -p30 l 24 : What is thus the importance of gamma in a model then ?
- -p32-l18 : Why is it problematic ?
- -p37 l5-10 : This could have been introduced at the beginning!
- -p38 l21 : recall what is alpha

2) Analysis :

The second companion paper present results from an experimental study of particle travel distances down a slope, launched by a catapult system. Data is compared to previous experiments and field studies in an exhaustive manner and tested again the theoretical elements provided in the first companion paper (e.g. the expected Pareto distribution of travel distances). Data is well presented and well detailed so that I believe the 2nd study can be published within minor changes.

First, I do not exactly see why high speed imaging is used apart from determining launched velocity. Indeed, all the results shown in Figures present travel distances that can all be determined without video.

Second, I am not sure to understand how the Pareto fits to the experimental distributions are obtained : by fitting the Pareto parameters, or by estimating them independently with high speed imaging (such as the \beta\_z collision restitution parameter) ? I believe the theory would prove very robust if all parameters could be estimated independently via imaging (or other technique). This point is not clear enough and I would suggest the authors to clarify this while presenting their experiments.

Third, it is somewhat disappointing not to see any particle trajectory plotted, that would show the 'heating' (acceleration) for steep slopes, or 'cooling' for milder slopes. I believe much information can be extracted from an acceleration / velocity diagram, as was done for bedload transport in the authors' 2012 paper serie.

**Other comments:**

Fig 9 and 10 (and maybe others) : recall what is \beta\_z in the caption so that each figure is understandable by itself.

**3) Entropy :**

The third paper has been the hardest for me to follow, because it touches concepts from statistical physics, that are less frequent in the earth science research community. For what I understood, the author claim to generalize the maximum entropy principle to several energy-based physical constraints. With this approach, they find similar Pareto distributions as with the varying deposition probability framework developed in companion paper 1. I believe this is an important result that goes far beyond particle motion on hillslopes, so that I am not convinced that associating this study as a companion paper is a judicious choice. In my opinion, proposing this study to a more physically sound readership journal than esurf would have a greater impact (Physical review ?). However, I rely on the editor's and other reviewers point on view for this.

**Other comments:**

p1 l15 "... that is heavy-tailed for net cooling and light tailed for net heating" Isn't it the other way around ?!(3) precise that A can be between -B and infinity ?(16) What is notation E[] for ? You have already used it for energy...

p19 l 7: What is Occam's razor ?

**4) Philosophy :**

The fourth paper present a general discussion on probabilistic approach to rarefied particle motions. It correctly points the generality of such approach, and shows how continuum equation of motion extend

(within some subtle extra terms) to ensemble average quantities or probability distributions, even when the instantaneous particle flux is strongly intermittent.

While I completely agree with this viewpoint, and I believe the paper has its importance for the community, I am not sure how this relates specifically to the hillslope motions. Indeed, the use of ensemble averaging/probabilistic description to describe rarefied gas, bedload, or avalanches, and the scale dependence of fluctuations, is a much more general discussion that could fit in a standalone study, with dedicated title. Indeed, the 4th papers format dilutes in my sense the distinct messages the authors convey. Nevertheless, if the editors and reviewers think the inclusion of this paper as a companion paper is justified I will not argue against this.

One minor comment is the following. The authors point 2 equivalent probabilistic viewpoints, the Fokker-Planck equation (the linearization of the master equation) and the maximum entropy approach, originating from statistical physics (they discussed in the 1st and 3rd companion paper). In the discussion, I would include a third way, the Poisson representation [1], which has the attracting characteristic of being exactly equivalent to the Master Equation, while leading to continuous, analytically tractable PDEs. This approach, developed by Gardiner, can be used [1,2] to compute the exact particle number pdf and correlations from basic entrainment/disentraiment rules, without requiring a "small" noise or Kramer-Moyal expansion that assume a large number of particles. As pointed by Gardiner, it has the potential to describe "low density-high fluctuations" states of granular gases, for which large deviations play an important role. A mention of such alternative could be relevant.

1 Gardiner, C. W. (1985). *Handbook of stochastic methods* (Vol. 3, pp. 2-20). Berlin: springer. 2 Ancey, C., & Heyman, J. (2014). A microstructural approach to bed load transport: mean behaviour and fluctuations of particle transport rates. *Journal of Fluid Mechanics*, 744, 129-168. 3 Heyman, J., Ma, H. B., Mettra, F., & Ancey, C. (2014). Spatial correlations in bed load transport: Evidence, importance, and modeling. *Journal of Geophysical Research: Earth Surface*, *119*(8), 1751-1767.

---

## Referee Comment (RC2) · Anonymous Referee #2 · 5 Feb 2021

Paper Entitled: Rarefied particle motions on hillslopes: 3. Entropy Author: D. Furbish, S. Williams, and T. Doane Manuscript no.: https://doi.org/10.5194/esurf-2020-100

Recommendations: This paper presents a theoretical analysis based on generalization of energy based constraints and maximum entropy method to study particle motions on hillslopes. The authors suggest that the generalized Pareto distribution is a maximum entropy distribution and represents the most probable arrangement of particles on a surface based on their travel distances. In general, the paper is interesting and well written, however, I feel, based on its extensive (or to a certain degree, entirely) mathematical and physics based content, it doesn't fit very well in a geomorphology journal. Also, it appears most of the discussion in on gaseous particles and little is related to the field or experimental observations of sediment particle on hillslopes. Given

the data and methodology employed, the article is relevant to the journal, but major changes and clarifications ought to be made prior to its publication.

Here are my specific comments:

-"The generalized Pareto distribution is a maximum entropy distribution. . .." I believe this is true for a distribution for a given sample size? If we compare tails of two distributions, e.g. exponential vs. power-law, both truncated, for example, due to finite size of the system (e.g. flume length etc.), would exponential have higher entropy (Shannon) than power-law?

Page 2 Line 20: What does energetic cost represents physically in terms of a moving particle? Could higher energetic cost be associated with shorter but more frequent waiting times for a moving particle? Is this energetic cost independent of particle size? - Can heating or cooling of particle be related to acceleration or deceleration of that particle moving on a hillslope?

Page 7 Line 8: Not clear why j is defined as $j = 0, 2, \ldots n$ (even) whereas later in the discussion $g_1(x)$ is computed.

Page 9 Line 26-30: Again, I think it would be easier for a reader if these discussions are written in a more accessible way to geomorphology community as it gets confusing while reading (e.g. use of terminology such as net cooling vs. net heating) whether the discussion is about gaseous/heat particles or sediment particles moving down slope.

Page 10 Line 8/Figure 4: ". . .increases more slowly with increasing distance x." What does it indicate? Does it imply tracer particles require less energy if they travel further/longer? If this is true, is it because of the achieved momentum? In that case, I assume larger particles will have higher momentum once entrained and require less energetic cost. On the other hand, what would happen to smaller particles if the waiting time is not sufficiently long enough; would they not require more energy to overcome resistance caused by trapping/hiding etc. to travel the same distance? With these

thoughts, I wonder, if this curve (Fig 4) is able to differentiate between particles coming from a wider GSD. Page 10 Line 9: Assuming tracer particles follow exponential distribution for travel distances; how would the hillslope surface look like? Would it be flat? Or in other words, if these particles were traveling on a river bed would we expect isothermal type of behavior for a plane bed conditions?

Page 13: Is there a range associated to energetic cost? What does it physically imply for a particle to have this cost as, for e.g., $w = 3$ vs $w = 10$? -Is this cost defined by the system size (e.g. hillslope length scale)? But shouldn't it also depend on resistance encountered by a particle while moving down slope?

-Looking at Figs. 4 and 5, for an isothermal process, can it be said that the maximum travel distance (as energetic cost and travel distance follow linear relation based on Fig 4) for a particle is 7 unit? Does it relate to maximum hillslope length?

Page 13 Line 10: So far it is not clear whether these curves (Fig 4/Fig 5) are for sediment particles or gaseous particles!

Page 1-Line 8: "the many different ways to arrange a great number of particles into distance states where each arrangement satisfies the same fixed total energetic cost - the generalized Pareto distribution represents the most probable arrangement."

- I assume these observations are for a constant slope?

- I wonder if authors have looked at the topographic fluctuations of the surface where these particles traveled to see if the particle distribution and their associated spatial arrangement are related to topographic fluctuations. Or in other words, can one infer the shape of the distribution from the topographic fluctuations as it is easier to obtain topographic data compared to travel distance distribution? Also, it would be useful for readers to show as an example spatial arrangement of particles based on Pareto vs exponential distribution.

In my opinion, this paper, in the current form, will fit well in a more physics based journal

such as Phys. Rev or Phys. of fluids etc.

Certain times the paper also appears as a review paper with several discussions (e.g. Pages 9, 16, 17) based on previous published/ in-review papers.

Also, I apologize for not reading the other three companion papers in case if I missed something; however, I feel that this paper should be standalone in a way that one should get most out of it while only reading it.

-Minor:

Page 1 Line 15 (or I would say the whole abstract): I think, perhaps it would more accessible to geomorphology community if this is written more from a perspective of tracer particle movement vs heating or cooling of particles.

---

## Author Comment (AC1) · 26 Mar 2021

**General response:** We appreciate the efforts of the referees in reviewing our work. Our revisions and responses below are based on careful consideration of their comments, questions and recommendations. As here, our responses below appear in blue font.

We note that the reviews of our work are mostly focused on the need for clarification. In response we have added a significant amount of material, particularly in the first companion paper. Nonetheless, these additions effectively represent minor revisions of the original manuscripts, as we have made no changes to technical elements of the work. We also have opted to make only some of the suggested stylistic changes. For these reasons we have submitted our revised manuscripts in the two-column *ESD* format, in part to avoid the displacement of figures to the end of the text that occurs with the one-column format, as noted by R. Glade in her review of the second manuscript.

All revisions in the manuscripts appear in blue font. In addition, we have posted this response (to all referee comments on all four manuscripts) with each of the revised manuscripts. This is to provide the full context — comments and responses — for the revisions in each.
* * *
**General comments concerning all papers**

**Referee #1 (Joris Heyman)**

These 4th companion papers are all very relevant for the different messages and new results they convey. I have fully enjoyed this tough but inspiring reading. I have no major comments to make, although, as explained in part 3 and 4, I suggest submitting the last two studies separately (in esurf or other journal), since their scope is much more general than the hill-slope problem.

Yes! The scope of the material in the third and fourth papers and their relevance to topics in sediment transport *are* far more general than the problem of rarefied particle motions on hillslopes. However, by design the description of this specific problem in the first two papers provides a very concrete example — a clear launching point — for (re)introducing and elaborating the general topic of maximum entropy as applied to sediment transport. That is, in addition to clarifying the (probabilistic) mechanical basis of the different forms of the generalized Pareto distribution of particle travel distances, the third paper provides a great way to help fold the idea of maximum entropy into the more general conversation of describing the probabilistic physics of particle motions — whether involving transport on hillslopes or in rivers — starting from a clear example. Similarly, the description of the hillslope problem provides a concrete example to motivate the exercise (fourth paper) of stepping back to examine the more general topic of probabilistic descriptions of sediment transport — which to date have been disproportionately focused on river sediment — drawing "attention to commonalities in the formalism used to describe transport in different settings." In this spirit, the series has been specifically crafted for the readership of *Earth Surface Dynamics*.

We also appreciate the open access model of *Earth Surface Dynamics*. Setting aside the issue of intellectual property rights, our work was publicly funded. The public, including members of the scientific community, should have free access to the results of our efforts.

While pleasant, the writing style contains many didactic sidebars, anecdotes or humor that do not ease the understanding of an already complex message. I sometimes felt more like reading a

book than a journal article (the 4th papers format do not help concision neither). Beside precise structural points (see part 1 of the review), I would tend to think that it is possible to globally shorten the text, summarizing the ides, without altering the important results and transferring extra materials in supplementary material.

Yes, we admit that the presentation at times reads more like a book than a journal article. And we certainly appreciate the sentiment expressed — that there might be value in condensing the presentation of the material. Nonetheless, we have the strong sense — correct or incorrect — that much of the material presented in the series is unfamiliar to a significant proportion of the readership of *Earth Surface Dynamics*. For this reason we opted to provide more detail — with explanatory sidebar material (and appendixes) — than we might normally be inclined to include. The humor, of course, is intended to break things up during a difficult read, although we admit that we also enjoy being a little goofy while writing with seriousness about technical material.
* * *
**Rarefied particle motions on hillslopes: 1. Theory**

**Referee #1 (Joris Heyman)**

This first companion paper is the master piece of the serie, presenting all theoretical developments.

**State of the art** The literature review has been placed after the theoretical developments (Section 5 Related formulation), which, in my opinion, do not help to globally envision the originality of the proposed formulation with respect to existing ones, and understand the main challenges of the hillslope problem. I would suggest the authors to better highlight the originality of their approach based on a succinct literature review from the very beginning. This could also help to introduce the important variables.

We have added material pertaining to previous work in the Introduction, consistent with this recommendation and that of Referee #2 (R. Glade). However, we prefer our approach of presenting the basis and implications of previous work (Section 5) after fully developing the ideas of our work — in order to provide context for our quite specific comparisons with this previous work. This approach is common in the literature. Note that we also briefly preview probabilistic formulations of disentrainment, and the formulation of Kirkby and Statham (1975), at the end of Section 2.1 in view of the context provided by this section.

**Summary of findings** In addition to this originality statement, I believe that a simple summary of findings should precede the detailed theoretical developments. In contrast to the book format, we expect in a journal article to have a rapid understanding of the main results. I had to wait for the summary provided in the second companion paper to make me a clear mental image of the main ingredients of the theory proposed, which I have expressed this way:
1. Particle Mass conservation $dN/dx = - N/E_a$
2. The variation of the ensemble average energy is constant (since forces are constant ?): $dE_a/dx = \text{Cst} \rightarrow E_a = Ax+B$
Thus, the mean disentrainment rate is $P=- 1/N \, dN/dx = 1/(Ax+B)$, and the PDF of travel distances is a Pareto distribution, in place of the classical Exponential distribution found when P is a constant. Such ultra-simplified preamble would ease a lot the navigation into the details of the

theory latter on.

This is a useful idea. We have added a new Section 2.3 with this material, which also is consistent with the general recommendations of R. Glade concerning "road map" material.

**Terminology** I understand the analogy between statistical physics of gas and motion of particles down a slope, although I am a bit skeptical on translating all the technical vocabulary for this situation. For instance, the terms "*thermal collapse*", "*iso-thermal*" and "*net heating*" are not fully transparent with respect to gravity driven motions, and will remain obscure for a majority of readers. In my opinion the notion of "heat" in a gas refers to zero-mean velocity fluctuations, and is thus not perfectly suited to describe a net shift of mean velocities as is the case in non-equilibrium particle motion driven by gravity. I understand the authors conceive the thermal collapse as a net decrease of particle energy and the heating as a net increase of particle energy. However, if they would extend their statistical formulation to the evolution of higher statistical moments of energy states, there will be a confusion between drift (mean velocity) and diffusion (fluctuations around the mean). My suggestion would be to simply use the transparent terms of mean "deceleration" and "acceleration" of particles ? One of the drawback of using energy balance instead of mass and momentum conservation is that well defined (and measurable) variables such as particle velocity and acceleration are lumped into an energy state, which is less tangible to the observer. Then, it is very easy to understand the disentrainment rate in terms of a decelerating particle (disentrainment probability growing with x, A>0) or accelerating motion (disentrainment probability decreasing with x, A<0).

This is interesting! Our short response is this: Because of the compelling connection with granular gases — albeit involving distinct differences — we prefer to retain the current terminology and descriptions centered on energy conversions. Indeed, one of our objectives is to steer the conversation surrounding the problem of rarefied particle behavior toward the framework provided by granular gas theory (e.g., Brilliantov and Pöschel, 2004) while acknowledging and accounting for the special circumstances of rarefied particle motions on hillslopes. Our reasoning follows.

We start with several key points regarding granular gases, noting both similarities and differences between "normal" conditions and the rarefied conditions of particle motions on hillslopes.

As a reminder, the rarefied conditions that we describe do not involve particle-particle collisions, only particle-surface collisions. Indeed, the Knudsen number in any realization is effectively infinite. As described in the text and elaborated in Appendix B, the distribution $n_{E_p}(E_p, x)$ of energy states $E_p$ of the particle cohort (ensemble) varies with position $x$. Because the moments of this distribution are assumed to be defined, we could in fact also define the distribution of downslope velocities, thence the mean velocity and fluctuations about the mean. That is, we could formally define a granular temperature (Goldhirsch, 2008) and then associate this temperature with a granular internal energy content at any position $x$. But this is where the analogy with a normal non-equilibrium granular gas ends. The granular temperature thus defined for the rarefied problem is not physically relevant to this problem, and a granular internal energy does not physically exist. Indeed, granular energy is neither advected nor diffused in the sense of a normal granular gas system, for example, a granular flow. (Note that we purposefully avoid any reference to a granular temperature in relation to particle motions on hillslopes.) Moreover, quantities such as the granular density and pressure do not exist. In short, there are no "internal" gas dynamics whatsoever, as the rarefied conditions do not represent a particle system that evolves dynamically over time and space, as

with a granular gas in a box or in a conduit or over an inclined surface, each of which involves dissipative particle-particle collisions during the gas evolution. Yet the description of the spatial evolution of the distribution of the energy states of the particle ensemble remains entirely relevant. As mentioned in the text, the rarefied case represents a highly unusual — if not unique — granular gas. In fact, to our knowledge this particular granular gas problem has not been examined before. The closest direct analogue seems to be that reported by Almazán et al. (2017), who, building from the work of Volfson et al. (2006), show that the formalism used in describing the cooling and thermal collapse of a granular gas is akin to the formalism used in describing the dissipative energetics of a single nonelastic ball bouncing on a smooth horizontal surface (without energy input from vibration or gravitational heating).

Whereas the dynamics of an ordinary granular gas are centered on dissipative particle-particle collisions, in our problem the dynamics are centered on dissipative particle-surface collisions. This dynamic is fundamentally a boundary related phenomenon, not an internal one. In a standard granular gas, energy dissipation occurs during particle-particle collisions. But note that, whereas a dissipative collision between two particles generally leads to an overall loss of kinetic energy, the kinetic energy of one of the two particles may actually increase. In contrast, in our problem involving only particle-surface collisions, essentially all collisions involve extraction (dissipation) of the particle kinetic energy defined with respect to downslope motion. Thus, all collisions are "cooling" in the sense of reducing particle kinetic energy. Similarly, gravity provides a uniform "heating" in the sense of increasing kinetic energy, regardless of the particle energy state. Thus, we prefer to retain our ideas of cooling and heating without reference to fluctuating motions (and granular temperature), where cooling simply refers to the idea that kinetic energy is extracted from the particle cohort via collisional friction and heating refers to the idea that kinetic energy is added to the cohort via conversion of gravitational potential energy into kinetic form. The idea of thermal collapse then is entirely satisfactory (e.g., Volfson et al., 2006).

We certainly intend to leave open the possibility of moving toward a terminology that is closer to the ideas surrounding the role of granular temperature as used in standard granular gas theory — notably if we further unfold the theory in relation to the nonequilibrium evolution of the distribution $n_{E_p}(E_p, x)$ and its moments, or begin to explore finite Knudsen number conditions. Nonetheless, we reemphasize the point that our formulation involves the evolution of $n_{E_p}(E_p, x)$ with respect to space, whereas standard granular gas theory typically involves hydrodynamic-like descriptions of quantities such as the granular density, temperature, pressure and velocity that evolve with respect to time (in an Eulerian manner), where it is assumed that local continuum-like definitions of these quantities exist (e.g., Goldhirsch, 2008) — conditions that are not relevant in the rarefied problem that we examine.

Please note that we have added material to Appendix B to clarify the points above, which nicely follow from the first part of this appendix.

Fokker-Planck equation I understand the authors objective to cast their analysis into a fully probabilistic framework, although I did not get the necessity here to derive a complete Fokker-Planck equation for E if none of the higher moment are used latter on. Indeed, the authors introduce beta$^2$ (diffusivity of the energy state), which is never used afterwards. Why ? In my opinion, the shape of the pdf (Pareto) is only dependent upon the evolution of the disentrainment rate probability, not on the FP description of energy states. This is a 'simple' non-homogeneous Poisson process. Introducing the FP formulation is thus somewhat confusing for the main message. If this FP equation had an importance for the description of the difference between harmonic or algebraic average of the energy states (Ea, Eh), it might have been preferable to introduce this concept differently (I personally did not get this distinction entirely).

The phrase above highlighted in red states the essence of our reasons for presenting the formulation of the Fokker-Planck equation. More specifically, yes, deposition is an inhomogeneous Poisson process. But the rate of this process fundamentally depends on the particle energy state. So to get the deposition right requires getting the energy balance right. The Fokker-Planck equation describes the spatial evolution of the probability density function of particle energy states for varying hillslope conditions (e.g., the Kirkby number). In turn, this sets the values of the shape and scale parameters $A$ and $B$ in the distribution of particle travel distances. That the deposition rate and energy balance are coupled is succinctly illustrated by Eq. (64) and Eq. (65), and by the finite-difference equations, Eq. (96) and Eq. (97), which must be solved recursively for nonuniform hillslope conditions.

Indeed, the second-order term in the Fokker-Planck equation is not used in the final description of the energy balance. But this is because of our assumption that the associated Péclét number is large, which is a standard type of argument in this type of analysis. We prefer to explicitly show where the energy balance comes from in relation to the probability distribution of energy states as this distribution changes with downslope position $x$. Then, the harmonic average energy naturally enters into the analysis — which is essential for revealing how deposition influences the energy balance via the preferential culling of low energy particles, analogous to the effects of aggregation described by Brilliantov et al. (2018).

**Meta-stability :** Being familiar to the study of Quartier et al. 2000, I wondered if the theoretical description proposed by the authors is also able to explain the occurrence of meta-stable states of motion due to micro- roughness. Indeed, depending on the initial particle velocity, a particle may be trapped by bed roughness or continue its motion indefinitely. I would have liked to find a mention of this somewhere in the text.

We describe elements of this paper in Appendix J, including its similarity with our work in relation to particle energy extraction, that is, as described by Eq. (4) in Quartier et al. (2000). But because the formulation and experiments involved are so different from ours — the restricted degrees of freedom of motion, the periodic (versus random) roughness elements, the continuous (versus discontinuous) contact between the moving particle and the rough surface, the mostly deterministic (versus probabilistic) qualities of the analysis — we prefer to not attempt to map our description onto the behavior described by Quartier et al. (2000) beyond what we offer in Appendix J. Even with large Kirkby number and large initial energy (velocity), our formulation suggests that there is still a finite probability that deposition will occur (except in the unrealistic limit that the deposition length scale goes to infinity). Nonetheless we have added material to clarify that in the experiments of Quartier et al. (2000) the "condition of a constant roller velocity involving an "equilibrium between gravity driving and dissipation by the shocks" is roughly analogous to isothermal conditions described in the main text, but without effects of deposition." Moreover, "the dynamical angle in these experiments coincides with the situation in which the velocity of the roller is sufficient to prevent trapping, "assuming a permanent contact between the roller and the rough plane."" We also further highlight in the main text that Gabet and Mendoza (2012) made use of the idea of collisional friction introduced in this paper, and that our energy extraction quantity $\beta_x$ is akin to the dissipation factor introduced therein.

Quartier, L., et al. "Dynamics of a grain on a sandpile model." *Physical Review E* 62.6 (2000): 8299.

Specific Points :

- p5 l8 : I did not get in which sense these probabilistic formulation are "scale independent"

This point has been elaborated previously, and we have added appropriate references and wording. In short, there are no length constraints imposed on the probability density $f_r(r; x)$.

- p5 l17 : "can be a **constant** determined"

Good catch. We have modified the wording.

- p8 l9 : "The law of the unconscious statistician" ...which means for an unconscious reader ?

LOL. Albeit a bit snarky, LOTUS is indeed an accurate definition. See for example: https://en.wikipedia.org/wiki/Law_of_the_unconscious_statistician
On this webpage the paper by DeGroot et al. (2014) offers an explanation, and here is a link that offers a bit of history:
https://math.stackexchange.com/questions/1500751/the-law-of-the-unconscious-statistician

- p9 l15: This sidebar could come before, at the beginning of the section

Agreed. We moved this sidebar, which is consistent with the recommendation (R. Glade) to add introductory material.

- p10 l20 : "So bear with us". This do not presage good...

LOL. Agreed.

- p11 l25 : Think of moving this didactic sidebar in annexe

We prefer to retain this sidebar at this location for the reasons offered above in relation to the familiarity of the material.

- p 12 l 23 : What does "immaterial" mean in this context ?

To mean unimportant or irrelevant. We prefer the preciseness of this word.

-p15 : The authors mention "deposition" in granular gases. I do not understand well how particles can deposit in absence of boundary. Do the authors mean "aggregation"?

We do mean deposition rather than aggregation (although aggregation is well studied in granular gas dynamics). Indeed, deposition does require the presence of a boundary. For example, see Volfson et al. (2006), Kachuck and Voth (2013) and Almazán (2017).

-p16 (39) and (40) : beta and beta$^2$ have the same units ?

Yes, both are dimensionless.

-p19 l6 "disentrainment rate, consistent with the deposition rate." I do not understand this.

We have added wording to clarify this point.

-p20 l25-30 This paragraph is very confusing for me. Could you reformulate it in simpler way ?

We reworded and simplified parts of this paragraph, and removed a nonessential sentence.

-p28 l17 **m** g mu cos theta

Ouch. That's embarrassing. Fixed.

-p30 l 24 : What is thus the importance of gamma in a model then ?

The effect of deposition on the energy balance vanishes if the term involving $\gamma$ is not present.

-p32-l18 : Why is it problematic ?

We have reworded the sentence to clarify this point.

-p37 l5-10 : This could have been introduced at the beginning!

In effect this material is presented at the beginning — starting with the Abstract — although we then introduce its elements systematically so that the summary here follows from the full context provided (e.g., the full meaning of the Kirkby number).

We have added material in the Introduction.

-p38 l21 : recall what is alpha

We have added wording, noting that this quantity is elaborated in the next sentence with reference to Eq. (51).

**Referee #2 (Rachel Glade)**

Furbish et al. present a novel theoretical analysis of hillslope sediment transport in steep landscapes. They carefully and thoughtfully lay out the pieces of the problem mathematically, incorporating both probabilistic and physical elements of the rarefied(non-continuum) transport of particles. They link their findings to concepts in granular gas theory, including both fundamental and recent discoveries in that field. This paper represents a substantial step forward in the field of geomorphology, with implications not only for hillslopes but for other environments as well. I believe their precise, careful approach will stand as an example for future theoretical studies in geomorphology.

As far as I can tell the math in the study is sound, and I appreciate the care with which each equation is presented and explained. The paper is generally enjoyable to read, and I appreciate the easter eggs and asides hidden throughout. That said, I think the paper could be lightly restructured to better highlight 1) where this study fits into previous work 2) testable predictions arising from theory. These two recommendations serve multiple purposes, the primary goal is to increase the readability of the manuscript and allow readers to better grasp the novelty/practicality and implications of the work in a coherent way.

Previous work and background: The authors could do a better job of first explaining (briefly) the state of the field before launching into their theory development. The first paragraph of the paper is nice, and does a good job of briefly highlighting previous studies on non-local transport. I think a new second paragraph could better lay out the fundamental problem: what does "rarefied" mean and why is it necessary to take into account rarefied transport on hillslopes? Defining the Knudsen number could be of use in this early part of the paper. This is a relatively minor change, but I think it would greatly improve the readability of the paper, especially for those who are not already well-versed in the concept of rarefied motions.

We have added material to this section, consistent with this recommendation and those of Heyman and Haff (see fourth paper below).

Testable predictions/theory overview: The main text currently stands at 98 equations(not including the appendices!). I strongly suggest the authors include a summary of the key equations in the discussion/conclusions section of the paper, perhaps in a table. Along with this summary, it would be helpful for the discussion to highlight key predictions testable in experiments or the field (this can be brief). This will help bring the reader full circle to remember what the theory aims to describe, and will also help connect this paper with the second paper, which tests theoretical predictions.

Because our equations are not like "benchmark" formulae that one might expect to see presented as such in an engineering review paper, nor of import like the celebrated equations of physics (https://cosmosmagazine.com/physics/six-physics-equations-changed-course-history/), we prefer to avoid a table. Note, however, that we purposefully highlighted the mean travel distance, Eq. (83), as it contains all key quantities influencing particle motions — thereby serving as a reference point for our discussion of the effects of these quantities, including our uncertainty about them. The paper presents ideas (highlighted with context in the Discussions and conclusions) that might be examined in laboratory and field settings, but its purpose is to present a theory of particle motions rather than to outline things that can be done in the laboratory or field. This is the purpose of the second companion paper.

Other comments: This paper has many moving parts. Consider adding a figure early in the paper that visually defines the main pieces of your analysis in the context of hillslopes- heating, cooling, etc. The parts could be referenced in the various sections of the paper that deal with each aspect of the theory.

We prefer to not add a figure early in the paper that attempts to illustrate things such as particle heating and cooling before having mathematically defined (with supporting explanation) the key elements that make up these things. We trust that readers can visualize particles moving down a

rough inclined surface due to gravity, intermittently colliding with the surface — perhaps inspired by Figure 1. (Figure 5 only makes sense because the reader by this point presumably has a clear idea of the intended meaning of heating and cooling.)

It would be helpful to have a summary sentence at the top of each section reminding the reader of where we are throughout the paper. Linking these with an overarching figure as describe above would also help.

We have added material at various locations.

Abstract You might consider starting with a more visual sentence about boulders/sediment rolling down steep hills - might help draw the reader in and parse the rest of this fairly technical abstract.

Whereas we appreciate the sentiment of this recommendation, we prefer the current opening sentence, as it accurately and succinctly states what the entirety of the paper is about.

Page 4 Line 9 only on hillslopes, or in any system?

We have added much material that includes clarification of this point.

Page 4 line 16 Can you very briefly explain what survival function means physically, as you do for $f_r$?

As stated in the sentence, it is the same as the exceedance probability. The sentence appearing below Eq. (3) with "survived" then provides additional physical interpretation.

Figure 2 This figure is very helpful. Consider moving it to section 2.

This specifically serves as a definition diagram for the material in Section 3 once we have set $r \to x$.

Page 7 line 11 Change "treat" to "treating"

Changed.

Page 8 line 6, 7 Change "Becomes" to "become." Either way is technically correct but the latter sounds better.

This refers to "number," so we prefer "becomes."

Page 8 line 19 What is transport of energy in this context?

Transport implies movement over space. Nonetheless, we have added clarification.

Page 12 line 25 Does total energy actually increase? Or just kinetic energy?

$E(x)$ denotes total kinetic energy. We have added the word "kinetic."

Page 15 line 3 "k3" should be "kd"

Oops! Fixed.

Page 16 line 7 "defined below." below or above? I thought we just read about this.

True, but this term has not yet been defined explicitly. We added wording.

Figure 3 It's a little difficult to understand how to read this figure. Do the arrows indicate a translation of the uniform distribution to the right or left? If so, then it is difficult to understand the triangular region in part C (even with the explanation in the caption). Upon re-reading I think I understand part C. Does the triangular region make part C no longer a uniform distribution? This might help the read understand...

We have revised the caption of this figure to clarify its meaning, consistent with the explanation in the accompanying text.

Page 20 line 24 "deposition occurs..." is this because energy is being added to the system?

No, deposition is balanced with entrainment.

Page 21 line 2 This thought experiment feels unfinished. What should we take away from it regarding hillslopes?

Consistent with this recommendation and that of Heyman, we have slightly reworded and simplified this paragraph, and added a sentence to reinforce its later relevance.

Page 22 line 5-7 This connection to a recent discovery in granular gases is really neat. Can you add a bit about how it impacts hillslope transport explicitly?

We agree that this recent discovery is particularly interesting. Nonetheless, we prefer to not elaborate this idea here. We cited this paper by Brilliantov et al. (2018) only to highlight the idea that heating can occur with aggregation in a "standard" granular gas, analogous to our finding that an apparent heating occurs with deposition — which influences the energy balance as described with reference to Eq. (57).

Page 23 line 1 Add a couple sentences of intro explaining what this section means and where were going now

We have added material.

**Addendum:** We have modified the caption of Figure 4 so that the notation is now consistent with the figure and text.

DJF, JJR, THD, DLR and SGWW (on behalf of AMA)
March 2021
* * *
**Rarefied particle motions on hillslopes: 2. Analysis**

**Referee #1 (Joris Heyman)**

The second companion paper present results from an experimental study of particle travel distances down a slope, launched by a catapult system. Data is compared to previous experiments and field studies in an exhaustive manner and tested again the theoretical elements provided in the first companion paper (e.g. the expected Pareto distribution of travel distances). Data is well presented and well detailed so that I believe the $2^{\text{nd}}$ study can be published within minor changes.

First, I do not exactly see why high speed imaging is used apart from determining launched velocity. Indeed, all the results shown in Figures present travel distances that can all be determined without video.

Yes, we determined particle launch velocities from high-speed imaging. But in addition, all drop-rebound motions leading to the data in Figure 9, Figure 10 and Table 4 are based on high-speed imaging. And, our qualitative description in Section 3.3.2 of particle-surface collisions involved examining high-speed imagery, examples of which appear in the Supplementary Materials (Vanderbilt University Institutional Repository, https://ir.vanderbilt.edu/handle/1803/9742).

Second, I am not sure to understand how the Pareto fits to the experimental distributions are obtained : by fitting the Pareto parameters, or by estimating them independently with high speed imaging (such as the $\beta_z$ collision restitution parameter) ? I believe the theory would prove very robust if all parameters could be estimated independently via imaging (or other technique). This point is not clear enough and I would suggest the authors to clarify this while presenting their experiments.

The fitting of the generalized Pareto distribution is fully described in the lengthy paragraph at the end of Section 2.3, which in turn points to a rather lengthy supporting appendix (Appendix A: Parameter estimation). As described, we fit the shape and scale parameters $A$ and $B$ directly from the data, then used these to constrain the values of $\mu$, $\alpha$ and $Ki$. The eventual goal would be to refine the theory in a manner where these quantities can be related to hillslope surface conditions (e.g., roughness) in order to then constrain $A$ and $B$. But this currently is not possible. In this regard, among other things Appendix A examines the pros and cons of one popular estimation method, the maximum likelihood method, which can be problematic when applied to the generalized Pareto distribution. To this we have added comments on the method of moments, which, although appealing, is unusable with censored data and when either the first or second moment is undefined.

Third, it is somewhat disappointing not to see any particle trajectory plotted, that would show the 'heating' (acceleration) for steep slopes, or 'cooling' for milder slopes. I believe much information can be extracted from an acceleration / velocity diagram, as was done for bedload transport in the authors' 2012 paper serie.

Our experimental setup is inadequate for doing full trajectories with high-speed imaging. That said, we are planning related experiments for which our experimental setup with high-speed imaging is adequate.

Other comments:
Fig 9 and 10 (and maybe others) : recall what is $\beta_z$ in the caption so that each figure is understandable by itself.

We have added wording to the captions.

**Referee #2 (Rachel Glade)**

In part 2 of the Rarefied paper series, Furbish et al. analyze a combination of previously published field and experimental data, as well as new experimental data, to compare with predictions from the rarefied hillslope sediment transport theory independently developed in the first paper. They first summarize the key theoretical components and predictions from the first paper. I found this section very helpful, as it crystallizes key aspects of the first paper. Next, they step through a series of experimental and field studies to examine different aspects of the theory, and find that all data support their general predictions: particle travel distances can be characterized by a general Pareto distribution, where the specific form of the distribution is controlled by the kinetic energy balance of the particles. They include a useful discussion of limitations of their work, along with suggestions for future studies that can untangle some of the unknown details of particle behavior. The videos from the Vanderbilt experiments are delightful, and really help to visualize a lot of the concepts presented in the paper. I found their careful analysis convincing and mostly well-presented, and I imagine this paper will become canonical among those studying sediment transport not only on steep hillslopes, but in a variety of settings.

I have only minor comments for this paper. While much of the paper is clear and well-organized, a couple elements remain unclear: 1) the role of grain size/angularity and how it relates to theory 2) the difference between presented experiments and field studies, and why they test different aspects of the theory. To be clear, these points are both discussed extensively in the paper, but without organized explanations both toward the beginning of the paper and at the beginning of each new section, they are a bit hard to follow. One unclear point relates to the subtle difference between spherical particles traveling over a rough surface, and angular particles traveling over a smooth surface, with seemingly similar effects. I think this is one of the more interesting points of the paper, but it is currently lost without being set up properly in the introduction.

We are a bit puzzled by this. Yes, we do discuss effects of particle size and rounded versus angular particles throughout the paper. But none of the reported experiments involve spherical particles moving over rough surfaces. (Perhaps there was some confusion due to the fact that LaTeX moved Figures 11–19 to the end of the manuscript; see comments below.) The particle used by Gabet and Mendoza (2012) was nominally spherical with respect to its three axes, but it otherwise was sub-angular. We also used spherical particles (marbles) in the drop-and-rebound experiments. Just for fun we released marbles onto the sand-roughened surface described in the Vanderbilt experiments, but none stopped on the surface and we did not report this in the paper. We also note that because this paper is focused on particle travel distances, we purposefully avoided a detailed description of the effects of particle shape in anticipation of submitting a separate manuscript focused on this topic. This manuscript (Williams and Furbish, 2021) is now under review with *Earth Surface Dynamics*. We also note in the Introduction that, "outstanding questions concern how particle size and shape in concert with surface roughness influence the extraction of particle energy and the likelihood of deposition," and we describe why in the main text and in the Discussion and

Line by line comments:

Page 2, Lines 4-5: can you give an example of another system where this work might be relevant?

We have added an example.

Page 2, Lines 10-21: This is an excellent summary of the theory presented in the first paper.

Page 2, Lines 24-25: are you not mainly summarizing theory from the first paper?

Yes and no. Context for the theory does come from Furbish and Haff (2010) and Furbish and Roering (2013), so we are offering due credit.

Page 2, Line 28: change to "new laboratory experiments" to show that they are being reported for the first time in this study

Added.

Page 11, Line 23: "Section 3, Laboratory Measurements": Add a little intro here to remind us where we are. "Now we're going to summarize experimental studies and compare their results to our theory..." Also consider adding a very brief summary of the results for both experimental and field comparisons, as it will help guide the reader through the various points of comparison in the coming sections...

Because we state this as the objective of our paper in the Introduction, we prefer to assume readers are on board and thus avoid this unnecessary stylistic addition. After all, the headers are basically screaming, "OK! So now we're doing the experiments and data part of the paper!" :-)

Figure 3 and 4: What do the different symbols in the figure correspond to?

We have added descriptions of the symbol colors in the captions. Note that our reuse of individual colors is to highlight the differences in the data sets while minimizing our page charges.

Page 14 line 8: "bumpety bump" I sincerely hope this technical wording remains in the final version of the paper.

LOL. (DJF: I am reminded of a comment offered by a reviewer of my paper in *JGR-ES* with co-author Peter Haff. The title has the word 'divot' and the phrase 'divers length scales.' Regarding 'divers' (diverse, many) the reviewer said the last time he saw that word was when he read Chaucer, and that he hoped it survived the editorial police.)

Page 15 line 17: can you more explicitly state how these experiments differ from those of Gabet? What aspects of the theory will you be comparing for this set of experiments?

We have added a sentence.

Page 17: "Experiments" Summarize briefly (1 sentence) what aspects of theory you will be comparing. Did you also measure travel distances to be able to make a plot similar to figures 3 and 4? Oh, I see. Why are figures after figure 10 placed at the end of the manuscript? I'm sure this will be fixed in editing but it currently hides some of the most exciting results of the paper.

Wanting for space to fit 28 figures and 10 tables in this single-column format, LaTeX spills many of the figures to the end of the document. Apologies. It does fine in two-column format, and this is the format we have used for our revised manuscripts.

Vanderbilt Experiments: Though you test different grain sizes, it is unclear what effect this has in your experiments. Is there a plot you can show highlighting the effect? Or a short amount of discussion?

The effect of particle size (and shape) is explicitly illustrated by the data sets in Figure 11, with accompanying explanation offered in the referencing paragraphs.

Page 24, section 4.1: Typo. "DiBiase"

Oops. Fixed.

Page 26, line 27: did this experiment endanger helpless banana slugs? I sure hope not.

See next.

Page 26, line 31: Phew.

Indeed.

Page 30, lines 8-9: Why not plot the data like this to show us?

The entire data set is plotted together in one figure in both the third and fourth papers. With $N = 5671$ the individual data sets strongly overlap in these plots. Here we prefer to keep them separate to show more detail (domain, range, fit) for the individual sets.

Page 33, line 39: It's unclear what this means - can you expand on it a little?

We have added wording to clarify the point being made.

DJF, SGWW, DLR, THD and JJR
March 2021
* * *
**Rarefied particle motions on hillslopes: 3. Entropy**

**Referee #1 (Joris Heyman)**

The third paper has been the hardest for me to follow, because it touches concepts from statistical physics, that are less frequent in the earth science research community. For what I understood, the author claim to generalize the maximum entropy principle to several energy-based physical constraints. With this approach, they find similar Pareto distributions as with the varying deposition probability framework developed in companion paper 1. I believe this is an important result that goes far beyond particle motion on hillslopes, so that I am not convinced that associating this study as a companion paper is a judicious choice. In my opinion, proposing this study to a more physically sound readership journal than esurf would have a greater impact (Physical review ?). However, I rely on the editor's and other reviewers point on view for this.

Our work indeed represents a generalization of the maximum entropy principle to include the bounded and heavy-tailed form of the generalized Pareto distribution — although ours is not the first effort focused on heavy-tailed distributions. And, yes, the approach goes far beyond the problem of particle motions on hillslopes, with possible implications for describing other sediment transport problems. Nonetheless, as outlined in our response above (General comments), the analysis serves to clarify the different forms of the generalized Pareto distribution in the hillslope problem specifically, and in turn this problem serves as a launching point for (re)introducing and elaborating the general topic of maximum entropy as applied to sediment transport. (We are contemplating the idea of preparing a separate paper aimed at a Physical Review-like readership.)

Other comments:
p1 l15 " that is heavy-tailed for net cooling and light tailed for net heating" Isn't it the other way around ?!

Actually, this perhaps counter-intuitive result is correct. As described in Section 5.1, net cooling lead to energetic costs $w$ that become unbounded in approaching $x = B/|A|$ (Figure 4), so the distribution of costs $f_w(w)$ is heavy-tailed (Figure 5). Conversely, net heating leads to decreasing costs with increasing $x$, so the distribution of costs is light tailed.

(3) precise that A can be between -B and infinity ?

Actually, $A \in \Re$ and $B > 0$, as stated below this equation.

(16) What is notation E[] for ? You have already used it for energy...

We have added a phrase to clarify that the non-italic E[ ] with brackets denotes the expectation (expected value).

p19 l 7: What is Occam's razor ?

We have added wording to clarify this.

**Referee #2 (Anonymous)**

Recommendations: This paper presents a theoretical analysis based on generalization of energy based constraints and maximum entropy method to study particle motions on hillslopes. The authors suggest that the generalized Pareto distribution is a maximum entropy distribution and represents the most probable arrangement of particles on a surface based on their travel distances.

In general, the paper is interesting and well written, however, I feel, based on its extensive (or to a certain degree, entirely) mathematical and physics based content, it doesnt fit very well in a geomorphology journal. Also, it appears most of the discussion in on gaseous particles and little is related to the field or experimental observations of sediment particle on hillslopes. Given the data and methodology employed, the article is relevant to the journal, but major changes and clarifications ought to be made prior to its publication.

Concerning the first phrase highlighted above in red:

Judging from websites associated with the journal *Earth Surface Dynamics* (https://www.earth-surface-dynamics.net/, https://www.earth-surface-dynamics.net/about/aims_and_scope.html and https://en.wikipedia.org/wiki/Earth_Surface_Dynamics), we are confident that this journal encourages manuscripts centered on understanding how Earth surface processes work. We are equally confident in suggesting that describing how Earth surface processes work, with relevance to geomorphology, often involves using mathematics and physics.

The work reported in our paper is entirely motivated by an Earth surface process — rarefied sediment particle motions on hillslopes — and the developments in the paper are aimed entirely at the probabilistic physics of these motions. An Earth surface processes journal is precisely where this paper belongs. Indeed, we crafted this series of four papers specifically targeting *Earth Surface Dynamics* and its readership. We reject the referee's view that our paper is unsuitable for *Earth Surface Dynamics* because of its "extensive... mathematics and physics based content."

Concerning the second phrase highlighted above in red:

The paper is given entirely to sediment particle motions. Our reference to gas particles — briefly in Sections 1 and 2.2 — is merely aimed at providing a brief summary of the history, development, justification and implications of a maximum entropy distribution and the MaxEnt method, whether applied to gas particles or to sediment particles or to numerous other systems not involving particles. Everything else in the paper is focused on sediment particles. Moreover, the Introduction (Section 1) offers explicit reference to the previous companion paper describing sediment particle motions involving both field-based and lab-based experimental observations. Indeed, the first paragraph ends with:

> "As described in Furbish et al. (2020b), these varying forms of the generalized Pareto distribution are consistent with laboratory measurements of particle travel distances reported by Gabet and Mendoza (2012) and Furbish et al. (2020b), and with field-based measurements of travel distances reported by DiBiase et al. (2017) and Roth et al. (2020)."

And, virtually all of Section 2.1 is given to a description of sediment particle motions, highlighting field and experimental observations in the final paragraph with reference to Figure 3 whose caption is:

> "Plot of modified exceedance probability $R_*$ versus dimensionless travel distance $x_*$ and line with log-log slope of -1 for laboratory experiments described by Gabet and Mendoza (2012) (green) and Furbish et al. (2020b) (red) and field-based experiments described by DiBiase et al. (2017) (blue) and Roth et al. (2020) (black). Data for $A < 0$ fall to left of $x_* = 10^0 = 1$ with values in the tails represented by smaller values of $x_*$. Data

for $A > 0$ fall to the right of $x_* = 10^0 = 1$ with values in the tails represented by larger values of $x_*$. Total data number is $N = 5671$."

We hope it is apparent from these and many other elements of the paper that it is very much about sediment particle motions, informed by both field-based and lab-based experimental observations.

Here are my specific comments:

We note that these comments are mostly questions which, although interesting, do not offer a basis for the reviewer's conclusion that the paper is unsuitable for publication in *Earth Surface Dynamics*, nor do they indicate what form of "major changes" are recommended or why. Some of the questions are addressed in the companion papers. But addressing them in this paper would be an unnecessary distraction from its purpose of demonstrating the maximum entropy idea and its implications for interpreting the different forms of the distribution of sediment particle travel distances.

- "The generalized Pareto distribution is a maximum entropy distribution...." I believe this is true for a distribution for a given sample size? If we compare tails of two distributions, e.g. exponential vs. power-law, both truncated, for example, due to finite size of the system (e.g. flume length etc.), would exponential have higher entropy (Shannon) than power-law?

That the generalized Pareto distribution in this problem is a maximum entropy distribution is unrelated to sample size, as is evident from the definition of the differential entropy, Eq. (15). Moreover, the entropy of a distribution does not depend just on its tail. The entropy of a distribution that is censored is independent of this censorship unless the distribution is re-normalized to a new distribution form that entirely neglects values greater than the censoring value. Different distributions in general have different entropy values. But the entropy of a given distribution can vary with the values of its parameter(s).

Page 2 Line 20: What does energetic cost represents physically in terms of a moving particle? Could higher energetic cost be associated with shorter but more frequent waiting times for a moving particle? Is this energetic cost independent of particle size?

As stated in this introductory section, the energetic cost is "the total cumulative energy extracted by collisional friction per unit kinetic energy available during particle motions." This physically means that collisions extract particle kinetic energy, which is defined as an energetic cost. This idea is then elaborated in great detail in Section 3. The problem does not involve waiting times. Particle size may enter the problem via the parameter $\alpha$, as described in Section 2, as this influences the shape and scale parameters $A$ and $B$. This section also points out that the particle energy balance is formulated for "for a given particle size and shape." Effects of particle size, which do not directly bear on describing the energetic cost, are more thoroughly examined in the second companion paper, "Rarefied particle motions on hillslopes: 2. Analysis."

- Can heating or cooling of particle be related to acceleration or deceleration of that particle moving on a hillslope?

Yes. Heating is associated with acceleration (by definition) in the conversion of gravitational potential energy to kinetic energy, and cooling is associated with deceleration (by definition) due to

extraction of kinetic energy by particle-surface collisions, as described in the Abstract and Section 1, and more fully in Section 2. The first companion paper, "Rarefied particle motions on hillslopes: 1. Theory," provides a thorough description of this process.

Page 7 Line 8: Not clear why j is defined as j = 0, 2,..., n (even) whereas later in the discussion g1(x) is computed.

Oops! Thank you for catching this. We have added the 1.

Page 9 Line 26-30: Again, I think it would be easier for a reader if these discussions are written in a more accessible way to geomorphology community as it gets confusing while reading (e.g. use of terminology such as net cooling vs. net heating) whether the discussion is about gaseous/heat particles or sediment particles moving down slope.

As described in the Introduction and in Section 2, heating refers to the addition of particle kinetic energy via conversion of gravitational potential energy to kinetic form, and cooling refers to extraction of particle kinetic energy via particle-surface collisions. We do not know what "...in a more accessible way to geomorphology community..." implies, as we are confident that individuals in this community are familiar with ideas of mechanical energy and conversions of this energy between its various forms. This section builds directly from the formulation of the energy balance in Section 2, which is focused entirely on sediment particles.

Page 10 Line 8/Figure 4: "...increases more slowly with increasing distance x." What does it indicate? Does it imply tracer particles require less energy if they travel further/longer? If this is true, is it because of the achieved momentum? In that case, I assume larger particles will have higher momentum once entrained and require less energetic cost. On the other hand, what would happen to smaller particles if the waiting time is not sufficiently long enough; would they not require more energy to overcome resistance caused by trapping/hiding etc. to travel the same distance? With these thoughts, I wonder, if this curve (Fig 4) is able to differentiate between particles coming from a wider GSD.

Again, the problem does not involve particle waiting times. The formulation assumes a single particle size, but it can be generalized to many sizes as described in the second companion paper.

Page 10 Line 9: Assuming tracer particles follow exponential distribution for travel distances; how would the hillslope surface look like? Would it be flat? Or in other words, if these particles were traveling on a river bed would we expect isothermal type of behavior for a plane bed conditions?

Please note that our paper is not about tracer particles nor about particles traveling on a river bed. These topics are briefly addressed in the fourth companion paper, "Rarefied particle motions on hillslopes: 4. Philosophy," and do not bear on the maximum entropy distribution examined in this paper. Particle motions on a river bed involve fluid forces that are not relevant in this paper. We are pursuing separate work on the energetics of bed load particle motions. Initial analyses suggest that the form of the distribution of travel distances, which is not necessarily exponential in form, is determined by fluid forces acting on the particles in concert with particle-bed collisions.

Page 13: Is there a range associated to energetic cost? What does it physically imply for a particle to have this cost as, for e.g., w = 3 vs w =10?

As illustrated in Figure 5 with reference to Eq. (39), the domain of $w$ is unbounded, namely, "...the probability density $f_w(w)$ of energetic costs $w$ is unbounded (Figure 5)." The distributions plotted in this figure indicate that for any value of the shape parameter $A$, small energetic costs are more likely to occur than large energetic costs. The two paragraphs surrounding Eq. (23) and Eq. (24) fully explain the physical meaning of the energetic cost, specifically in terms of the kinetic energy extracted by collisional friction. This also bears on the following questions.

- Is this cost defined by the system size (e.g. hillslope length scale)? But shouldn't it also depend on resistance encountered by a particle while moving down slope?

No, the cost is not defined by system size, although a particle coming to rest at the base of a hillslope no longer experiences energetic loss. Yes, the cost depends on the collisional friction (resistance) encountered by the particle as it moves downslope, as described in Section 2 and Section 3.

- Looking at Figs. 4 and 5, for an isothermal process, can it be said that the maximum travel distance (as energetic cost and travel distance follow linear relation based on Fig 4) for a particle is 7 unit? Does it relate to maximum hillslope length?

These are unbounded functions for isothermal conditions ($A = 0$). The formulation is independent of hillslope length, although the maximum distance that a particle can travel on a real hillslope is limited by the hillslope length, assuming the base of the hillslope is a river channel, for example, rather than a flat surface at the base of a scarp or moraine. We do not understand the reference to "7 unit."

Page 13 Line 10: So far it is not clear whether these curves (Fig 4/Fig 5) are for sediment particles or gaseous particles!

The paragraphs surrounding these figures (and the associated references to the figures) are given entirely to sediment particle motions.

Page 1-Line 8: "the many different ways to arrange a great number of particles into distance states where each arrangement satisfies the same fixed total energetic cost - the generalized Pareto distribution represents the most probable arrangement." - I assume these observations are for a constant slope?

Yes, as described in Section 2. We prefer to not elaborate this detail in the Abstract.

- I wonder if authors have looked at the topographic fluctuations of the surface where these particles traveled to see if the particle distribution and their associated spatial arrangement are related to topographic fluctuations. Or in other words, can one infer the shape of the distribution from the topographic fluctuations as it is easier to obtain topographic data compared to travel distance distribution? Also, it would be useful for readers to show as an example spatial arrangement of particles based on Pareto vs exponential distribution.

We do not examine effects of topographic variations and roughness on travel distances in this paper. However, the first two papers in the series, "Rarefied particle motions on hillslopes: 1. Theory" and "Rarefied particle motions on hillslopes: 2. Analysis," do examine this question,

including reference to work on this specific topic reported by Roth et al. (2020). The forms of the generalized Pareto distribution, including its exponential form with $A = 0$, are shown in Figure 2. The same information is shown in collapsed form involving experimental data in Figure 3, so by inference these data are well represented by the various forms of the generalized Pareto distribution illustrated in Figure 2.

In my opinion, this paper, in the current form, will fit well in a more physics based journal such as Phys. Rev or Phys. of fluids etc.

Please see our comments above concerning this idea.

Certain times the paper also appears as a review paper with several discussions (e.g. Pages 9, 16, 17) based on previous published/ in-review papers.

Yes, we agree.

Also, I apologize for not reading the other three companion papers in case if I missed something; however, I feel that this paper should be standalone in a way that one should get most out of it while only reading it.

We both agree and disagree with this sentiment. In an ideal situation the material presented in a companion paper among several should be understandable without reference to the other papers. In this actual situation, however, to provide detailed context and answers to all possible questions arising from reading one of the papers would effectively require combining them. Each of the four papers focuses on a major, well-defined aspect of the problem of describing the probabilistic physics of rarefied particle motions. The subtitles of the four papers make clear what these topics are. In contrast, a single large paper containing four major topics — distinct but related — would risk "burying" these distinct topics in an unwieldy tome. For this reason we very purposefully devoted an entire section in each of the 2nd, 3rd and 4th papers to summarizing key material from the preceding papers. The material in each of these sections in each paper was chosen to provide essential material for understanding the presentation in the paper, but no more. We are appealing to the idea that an appreciation of the "full story" presented in companion papers might involve eventually reading all of them, although we appreciate that providing reviews for all four would be a burdensome task.

- Minor:
Page 1 Line 15 (or I would say the whole abstract): I think, perhaps it would more accessible to geomorphology community if this is written more from a perspective of tracer particle movement vs heating or cooling of particles.

We appreciate the referee's apparent interest in tracer particles, but the work reported in this paper is not directly concerned with tracer particles. The energetics involved with particle travel distances are part of the tracer problem. Another particularly important part involves describing the probabilities associated with particle entrainment on hillslopes — a largely unknown piece of the puzzle. We also note that only a small fraction of the geomorphology community is familiar with problems involving tracer particle motions. Please also see our responses above to Referee #1 (1. Theory) concerning our motivation for casting the problem in terms of particle energetics.

DJF, SGWW and THD
March 2021
* * *
**Rarefied particle motions on hillslopes: 4. Philosophy**

**Unsolicited Comment (Peter Haff)**

The present suite of papers focuses on the special case of rarefied grain collisions,where particle-surface interactions dominate mutual interactions between in-flight grains. Together these contributions illustrate the progression from micro-dynamics to macro-scale observation using laboratory and field-based measurements. This fourth paper on "philosophy" provides an extended discussion of the framework and strategy of the technical analysis, which is based on an analogy with statistical mechanics. I suspect that many geomorphologists who deal with sediment transport and landscape evolution, if they scan through recent ESD postings, will pause and poke only briefly,if at all, at this set of papers with its heavy reliance on concepts such as maximum entropy distribution and Fokker-Planck equations. In my view that would be a mistake. These papers open up many vistas on grain transport that might otherwise be missed.I would suggest the reader focus first on this final paper in the series. It alerts readers to a range of interesting problems that are difficult either to state or to resolve in the language of familiar continuum methods of analysis, but which can be usefully approached from a foundation of statistical mechanics. These might include problems in risk assessment for example, in which specific outcomes, even when many particles are involved, need not be reliably close to "average behavior" (the distinction made by the authors between granular "weather" and granular "climate"). Generic grain transport problems that might benefit by paying greater attention to statistics of the underlying particle dynamics include effects of granular size, shape, density, friction coefficient sand elastic moduli on erosion, sedimentation and sorting of particles. To aid accessibility and to draw readers in, it might be useful to add in abbreviated form some of the philosophical or "framing" content of the last paper into the early part of the text of the first paper of the series. This could help clarify at the outset the overall unity and import of the overall body of work, which I hope ESD will promote to full publication.

We have added material to the Introduction of the first paper in the series. This elaborates the content and overarching framework examined in the fourth paper, and adds further information regarding the significance of the results of the third paper.

**Referee #1 (Joris Heyman)**

The fourth paper present a general discussion on probabilistic approach to rarefied particle motions. It correctly points the generality of such approach, and shows how continuum equation of motion extend (within some subtle extra terms) to ensemble average quantities or probability distributions, even when the instantaneous particle flux is strongly intermittent.

While I completely agree with this viewpoint, and I believe the paper has its importance for the community, I am not sure how this relates specifically to the hillslope motions. Indeed, the use of ensemble averaging/probabilistic description to describe rarefied gas, bedload, or avalanches, and the scale dependence of fluctuations, is a much more general discussion that could fit in a standalone study, with dedicated title. Indeed, the 4th papers format dilutes in my sense the distinct messages the authors convey. Nevertheless, if the editors and reviewers think the inclusion of this

paper as a companion paper is justified I will not argue against this.

Please see our opening response to "General comments concerning all papers." We certainly agree that the probabilistic description is generally applicable to numerous topics. Nonetheless, for the reasons described in our opening response, this fourth paper is specifically crafted for the readership of *Earth Surface Dynamics* using the hillslope problem to motivate the material.

One minor comment is the following. The authors point 2 equivalent probabilistic viewpoints, the Fokker-Planck equation (the linearization of the master equation) and the maximum entropy approach, originating from statistical physics (they discussed in the 1st and 3rd companion paper). In the discussion, I would include a third way, the Poisson representation [1], which has the attracting characteristic of being exactly equivalent to the Master Equation, while leading to continuous, analytically tractable PDEs. This approach, developed by Gardiner, can be used [1,2] to compute the exact particle number pdf and correlations from basic entrainment/disentraiment rules, without requiring a "small" noise or Kramer-Moyal expansion that assume a large number of particles. As pointed by Gardiner, it has the potential to describe "low density-high fluctuations" states of granular gases, for which large deviations play an important role. A mention of such alternative could be relevant.

We have added a paragraph highlighting these points.

1 Gardiner, C. W. (1985). Handbook of stochastic methods (Vol. 3, pp. 2-20). Berlin: springer.
2 Ancey, C., & Heyman, J. (2014). A microstructural approach to bed load transport: mean behaviour and fluctuations of particle transport rates. Journal of Fluid Mechanics, 744, 129-168.
3 Heyman, J., Ma, H. B., Mettra, F., & Ancey, C. (2014). Spatial correlations in bed load transport: Evidence, importance, and modeling. Journal of Geophysical Research: Earth Surface, 119(8), 1751-1767.

**Referee #2 (Anonymous)**

In the 4th paper of the companion papers, the authors explain their scientific strategy and philosophical viewpoint for attacking the highly important and yet unsolved problem of soil transport on rough landscapes over a wide range of space and time scales. The paper covers a very broad range of significant issues, that all of them are pertinent to soil and sediment transport in various settings (and not necessarily on hillslopes). The work also sets a high bar and a valuable example for future research in this and adjacent fields. It further provides variety of ideas and problems that can become future research topics by other researchers and inspire much needed work to explore and illuminate the physics and mechanics of soil and sediment transport. At this point, I would like to add that the paper is also heavy in arguments based on statistical mechanics and kinetic theory of gases. This reviewer is not an expert in either of these fields, and they only have an introductory knowledge to follow the argument. Therefore, I encourage the editors or interested readers to further evaluate the statistical mechanics-based arguments presented in the paper by themselves, or by seeking additional input from experts in those fields (I noticed that editors might have already sought feedback from experts in those areas). I support the publication of this part of the companion papers, after the authors have revised the paper to address my mostly minor comments and questions below:

In considering the referee's comments below we recognize the need to reemphasize, early in our

presentation, that we are focused almost exclusively on rarefied conditions involving relatively rapid motions of particles over surfaces (the land surface or a streambed) rather than dense granular motions that occur, for example, in creeping soil. As stated in the Introduction, "the analyses of rarefied particle motions in these companion papers collectively provide an ideal case study for highlighting key elements of a statistical mechanics framework for describing sediment particle motions and transport." That is, the paper is purposefully *not* aimed at addressing all processes of sediment movement on hillslopes. We have added two key sentences in the Introduction, partly imported from the first paper in the series. These straightforward changes highlight our focus and largely address the questions raised by the referee below. Nonetheless, we offer responses in the spirit of providing further clarification — and because the topics are fun to think about!

This might have been discussed in the earlier parts of the companion papers — which I didn't have the resources to study in detail, before completing my review task — however, I would appreciate it if the authors can elaborate in this part, the conditions under which particle transport can be considered rarefied. I think providing a quantitative statement, potentially a dimensionless number or a metric that can be measured in experiment/simulation/field, would be very valuable.

Yes, this point is addressed in the first companion paper. Nonetheless, we agree that there is merit in adding clarification to this paper, and we have added a statement in the Introduction. As noted in the first paper, for an ordinary gas the onset of rarefied conditions occurs when the Knudsen number $Kn \geq 0.01$. However, this is not transferable to the situation of rarefied motions on hillslopes (or bed load), so our clarifying statement highlights the idea that effects of particle-surface interactions dominate over effects of particle-particle interactions in determining the behavior of the particles.

Related to my previous comment, do the authors think or advocate that all soil transport (or all that matters for soil transport phenomenon) is in rarefied condition? I have an opposing view, but at this point, it maybe just a matter of my misunderstanding. I try to explain my view in the next few sentences. I think of soil transport as a continuum — i.e., coexisting and gradually transitioning between them — of modes of transport and transport conditions. I agree that rarefied condition is a big part of that. However, as the ratio of particle per volume (packing fraction or solid fraction) increases as we get closer (from air) to (hillslope) surface, the transport condition becomes closer to the dense flow or dense particle transport. My view is in part informed by the experiments by Houssais et al (2015), where the authors show that in the case of fluvial sediment transport, there are three regimes (suspended particles, bedload, and creep; see their Fig. 1D). Some may argue that these are just two regimes by considering suspended transport as a part of the bedload, from the viewpoint of sediment transport, or by considering creep as part of the bedload, from the viewpoint of granular flow and physics; but in any case, there are at least two regimes there. How can the framework described here be applied to, or otherwise remain relevant to, such conditions in the lab and field, where the entire system may not be in the rarefied condition? Would you advocate for ignoring the contributions of the dense regimes, or otherwise suggest to readers to focus for future research on the rarefied condition of soil transport? Do you consider the contributions of the dense flow part of the transport as solved with the existing heuristic equations and relations, especially if they cannot be explained or adequately modeled in the rarefied framework described in the companion papers?

Delightful! No, we do not view relatively dense granular flows/transport as representing rarefied conditions. In the specific case of bed load, we view the relatively fast motions "at" the surface as

being akin to a rarefied gas (like particles moving over the surface of a hillslope), albeit strongly coupled with fluid motions. The relatively slow motions within the dense material beneath, giving way to granular creep with increasing depth in experiments (Houssais et al., 2015), do not represent rarefied conditions. And we note that this creep may contribute to "preconditioning" of the surface particles via particle rearrangements such that, for example, the onset of motions of surface particles by fluid forces and collective entrainment is not independent of deeper particle motions. In a sense, the processes of particle entrainment and deposition at the interface are akin to phase transitions between dense and rarefied states. But this analogy is not well developed and merits separate examination. Regarding soils, there are continuing questions as to the conditions under which granular materials (including soils) satisfy the continuum hypothesis in the sense of an ordinary fluid continuum, notably given the occurrence of behavior (e.g., shear focusing/banding, creation and relaxation of disordered structures including macro-pores) that cannot readily be described in a conventional continuum framework. We note that in his pioneering paper, Haff (1983) cautioned against assuming *de facto* that granular flows satisfy continuum conditions, and our impression is that granular physicists are careful in defining what they mean by a continuum in considering dense granular materials (e.g., Hennan and Hamrin, 2013). (This topic receives considerable attention in studies of granular gases, where clustering due to dissipative collisions is a strong part of the dynamics.) And, it is worth noting that there are efforts to show that kinetic theory, conventionally given to gases, also is relevant to dense granular material (Kim and Kamrin, 2020). Again with respect to soils, our sense is that recent efforts (e.g., Deshpande et al., 2021) are beginning to offer a clearer understanding of the important role of granular physics in soil creep. But this topic, too, merits separate examination.

The behavior of materials (and not general phenomenology of transport) in the dense regime is highly sensitive to size and size distribution, shape, etc of granular materials. I think there is an over emphasis in the manuscript on the probabilistic approach to the problem (which I feel like to be very useful for rarefied transport condition). However, to the degree that the contributions of dense flow regime remain relevant to soil transport, I would favor a slightly more balanced research and scientific strategy, where there is enough space to explore and investigate the physics and mechanics of dense regime. I would also argue for integrating more of reductionist studies, e.g., on the physics of mixing using the laboratory or simulation data on the feasible spatiotemporal scales, and less worry in the first instance about the uncertainty of those measurements for application to the geological space and time. The authors have mentioned this at some points in the paper (the part about discrete element modeling simulations or other first principle or ab initio studies that can be accompanied by carefully crafted experiments and theory development), but I think those avenues might be worthy of some more attention, especially in this philosophical part of the companion papers.

To be clear, we are fully on board with this type of effort. In effect this is the approach we take in the first two papers in the series (Theory, Analysis), albeit stated slightly differently in the Discussion and conclusions of this fourth paper:

> "In particular, this framework points us in the right direction for examining the physics of rarefied particle motions on hillslopes, wherein we see the behavior of the particle system precisely for what it is — an unusual granular gas. The effort then consists of elucidating a micro-view of the mechanical behavior of the particles during their downslope motions, which, when described probabilistically, leads to a macroscopic view of their collective (emergent) behavior."

To this we add that the approach is aimed at gaining mechanical insight as the basis for thinking about larger/longer scales. Although starting at small scale, this approach is not incompatible with putting uncertainty on the radar at the beginning. With respect to relatively dense granular materials, we note that there is now a significant effort aimed at describing the behavior of these materials using a statistical mechanics framework (e.g., Schröter and Daniels, 2012; Tejada, 2013; Bi et al., 2015). This framework is not necessarily aimed at uncertainty per se, but by definition it acknowledges the probabilistic qualities of the problem. We have added wording that acknowledges efforts focused on relatively dense granular motions, in parallel with efforts focused on granular gases.

This a very minor comment. On page 19, ∼line #30, the authors put close to each other, what is called "nonlocal behavior/rheology" in the dense granular flow research community, and the non-local transport in the sediment transport research community. First, this gives me the impression that the authors consider dense granular/particle flow to be in the rarefied condition. I don't think this is a correct view, but I am happy to learn more about the authors viewpoint on that, so it would be helpful if they can clarify on this matter. Second, I think there is still ongoing debate related to the dense granular flow behavior, and whether it should be called "nonlocal behavior" or "nonlocal rheology". This issue is not yet settled in the granular and complex fluids research communities, and the resembles between the two terms (nonlocal rheology and nonlocal transport) may cause confusion or concern for some readers.

As outlined above, we do not consider dense granular flows to be in a rarefied condition. Our reading of the granular physics literature suggests that researchers in this field usually are referring to nonlocal rheology (e.g., Hennan and Kamrin, 2014; Kim and Kamrin, 2020). Noting that "behavior" is generic, the key point of our reference to this idea in this and the first companion paper is to emphasize that, like descriptions of nonlocal granular rheology, we are describing a *physical* behavior in our use of the idea of "nonlocal" transport rather than defining local versus nonlocal transport based on a mathematical criterion (e.g., involving a light-tailed versus a heavy-tailed distribution of travel distances).

**References**

Houssais, M., Ortiz, C. P., Durian, D. J., & Jerolmack, D. J. (2015). Onset of sediment transport is a continuous transition driven by fluid shear and granular creep. Nature communications, 6(1), 1-8.

DJF and THD
March 2021
* * *
**Addendum:** In the first, second and fourth papers we have added material concerning efforts to treat the normal coefficient of restitution as a random variable (Gunkelmann et al., 2014; Serero et al., 2015) rather than a fixed quantity, analogous to our treatment of the energy extraction quantities, $\beta_z$ and $\beta_x$, as random variables.

**References**

Almazán, L., Serero, D., Salueña, C., and Pöschel, T.: Energy decay in a granular gas collapse, New Journal of Physics, 19 013001, 2017.

Bi, D., Henkes, S., Daniels, K. E., and Chakraborty, B.: The statistical physics of athermal materials, Annual Review of Condensed Matter Physics, 6, 63–83, 2015.

Brilliantov, N. V. and Pöschel, T.: Kinetic Theory of Granular Gases, Oxford University Press, New York, 2004.

Brilliantov, N. V., Formella, A., and Pöschel, T.: Increasing temperature of cooling granular gases, Nature Communications, 9, 797, doi: 10.1038/s41467-017-02803-7, 2018.

Deshpande, N. S., Furbish, D. J., Arratia, P. E., and Jerolmack, D. J.: The perpetual fragility of creeping hillslopes, https://doi.org/10.31223/osf.io/qc9jh, 2021. (in review)

DiBiase, R. A., Lamb, M. P., Ganti, V., and Booth, A. M.: Slope,vgrain size, and roughness controls on dry sediment transport and storage on steep hillslopes, Journal of Geophysical Research – Earth Surface, 122, 941–960, doi:10.1002/2016JF003970, 2017.

Furbish, D. J. and Haff, P. K.: From divots to swales: Hillslope sediment transport across divers length scales, Journal of Geophysical Research – Earth Surface, 115, F03001, doi: 10.1029/2009JF001576, 2010.

Furbish, D. J. and Roering, J. J.: Sediment disentrainment and the concept of local versus nonlocal transport on hillslopes, Journal of Geophysical Research – Earth Surface, 118, 1–16, doi: 10.1002/jgrf.20071, 2013.

Gabet, E. J. and Mendoza, M. K.: Particle transport over rough hillslope surfaces by dry ravel: Experiments and simulations with implications for nonlocal sediment flux, Journal of Geophysical Research – Earth Surface, 117, F01019, doi:10.1029/2011JF002229, 2012.

Goldhirsch, I. Introduction to granular temperature, Powder Technology, 182, 130–136, 2008.

Gunkelmann, N., Montaine, M., and Pöschel, T.: Stochastic behavior of the coefficient of normal restitution, Physical Review E, 89, 022205, 2014.

Haff, P. K.: Grain flow as a fluid-mechanical phenomenon, Journal of Fluid Mechanics, 134, 401–430, 1983.

Hennan D. L. and Kamrin, K.: A predictive, size-dependent continuum model for dense granular flows, Proceedings of the National Academy of Sciences, 110, 6730–6735, 2013.

Hennan, D. L. and Kamrin, K.: Continuum thermomechanics of the nonlocal granular rheology, International Journal of Plasticity, 60, 145–162, 2014.

Kachuck, S. B. and Voth, G. A.: Simulations of granular gravitational collapse, Physical Review E, 88, 062202, doi: 10.1103/PhysRevE.88.062202, 2013.

Kim, S. and Kamrin, K.: Power-law scaling in grnular rheology across flow geometries, Physical Review Letters, 125, 088002, 2020.

Kirkby, M. J. and Statham, I.: Stone movement and scree formation, The Journal of Geology, 83, 349–362, 1975.

Roth, D. L., Doane, T. H., Roering, J. J., Furbish, D. J., and Zettler-Mann, A.: Particle motion on burned and vegetated hillslopes, Proceedings of the National Academy of Sciences, www.pnas.org/cgi/doi/10.1073, 2020.

Serero, D., Gunkelmann, N., and Pöschel, T.: Hydrodynamics of binary mixtures of granular gases with stochastic coefficient of restitution, Journal of Fluid Mechanics, 781, 595–621, 2015.

Schröter, M. and Daniels, K. E.: Granular segregation in dense systems: the role of statistical mechanics and entropy, arXiv:1206.4101 [cond-mat.soft], 2012.

Tejada, I. G.: Aplicación de la física estadística a los medios granulares, Tesis doctoral, Universidad Politécnica de Madrid, 2013.

Volfson, D., Meerson, B., and Tsimring, L. S.: Thermal collapse of a granular gas under gravity, Physical Review E, 73, doi: 10.1103/PhysRevE.73.061305, 2006.

Williams, S. G. W. and Furbish, D. J.: Particle energy partitioning and transverse diffusion during rarefied travel on an experimental hillslope, Earth Surface Dynamics, https://doi.org/10.5194/esurf-2020-107, 2021.